# Contributions of anterior cingulate cortex and basolateral amygdala to decision confidence and learning under uncertainty

A. Stolyarova[1,10], M. Rakhshan [2,10], E.E. Hart[1], T.J. O'Dell[3,4], M.A.K. Peters[5,6,7], H. Lau [1,4,8,9], A. Soltani [2,11]* & A. Izquierdo [1,4,11]*

The subjective sense of certainty, or confidence, in ambiguous sensory cues can alter the interpretation of reward feedback and facilitate learning. We trained rats to report the orientation of ambiguous visual stimuli according to a spatial stimulus-response rule that must be learned. Following choice, rats could wait a self-timed delay for reward or initiate a new trial. Waiting times increase with discrimination accuracy, demonstrating that this measure can be used as a proxy for confidence. Chemogenetic silencing of BLA shortens waiting times overall whereas ACC inhibition renders waiting times insensitive to confidence-modulating attributes of visual stimuli, suggesting contribution of ACC but not BLA to confidence computations. Subsequent reversal learning is enhanced by confidence. Both ACC and BLA inhibition block this enhancement but via differential adjustments in learning strategies and consistent use of learned rules. Altogether, we demonstrate dissociable roles for ACC and BLA in transmitting confidence and learning under uncertainty.

[1] Department of Psychology, University of California, Los Angeles, Los Angeles, CA 90095, USA. [2] Department of Psychological and Brain Sciences, Dartmouth College, Hanover, NH 03755, USA. [3] Department of Physiology, University of California, Los Angeles, Los Angeles, CA 90095, USA. [4] The Brain Research Institute, University of California, Los Angeles, Los Angeles, CA 90095, USA. [5] Department of Bioengineering, University of California, Riverside, Riverside, CA 92521, USA. [6] Department of Psychology, University of California, Riverside, Riverside, CA 92521, USA. [7] Interdepartmental Graduate Program in Neuroscience, University of California, Riverside, Riverside, CA 92521, USA. [8] Department of Psychology, The University of Hong Kong, Pok Fu Lam, Hong Kong. [9] State Key Laboratory for Brain and Cognitive Sciences, The University of Hong Kong, Pok Fu Lam, Hong Kong. [10]These authors contributed equally: A. Stolyarova, M. Rakhshan. [11]These authors jointly supervised this work: A. Soltani, A. Izquierdo. *email: alireza.soltani@dartmouth.edu; aizquie@psych.ucla.edu

Learning relies on the ability to use external cues to predict the state of the world, take actions based on those predictions, and associate those actions with subsequent reward. Learning such associations can be straightforward when stimuli that precede actions or rewards can be discriminated clearly. However, this is not the case in naturalistic settings in which sensory cues or stimuli are ambiguous and thus the perception of or prediction about the state of the world is uncertain. In such situations, stimulus detection or discrimination are frequently accompanied by a sense of certainty, or confidence, in choice[1,2]. Recent evidence indicates that confidence may influence neural activity in brain regions involved in orchestrating reward responses[3,4] particularly when reward is significantly delayed[5]. Consequently, the sensory properties of reward-predicting cues and confidence in disambiguating them may directly influence valuation[6] and learning from reward feedback.

Recent studies in humans have revealed neural correlates of confidence estimation and learning in several brain regions, including the prefrontal cortex[7,8]. However, it is unclear whether these areas are causally involved in these processes. Despite powerful interference techniques in rodents[9], most rodents studies on neural mechanisms of confidence have been conducted within olfactory and auditory modalities[10,11]. In contrast, human studies on choice and learning under perceptual uncertainty have focused on visual processing, making it difficult to link findings across species.

Here, we trained rats to report the orientation of noisy Gabor patches by making spatial choices based on a learned stimulus–response rule (e.g., horizontal → left and vertical → right). We manipulated different aspects of the visual stimuli to alter performance and uncertainty associated with discriminating the orientation. Following action selection using a touchscreen, rats expressed their confidence by time-wagering: they could wait for a variable amount of time before they could receive a possible reward or initiate a new trial[11]. This design allowed us to measure confidence on a trial-by-trial basis. After ensuring that rats learned the stimulus–response associations, we reversed these associations to study the effect of confidence on learning. Extensive studies in rodents have shown a distributed network supports learning and choice involving uncertain outcomes, including basolateral amygdala (BLA)[12–14] and anterior cingulate cortex (ACC)[15,16]. Here, we use inhibitory designer receptors exclusively activated by designer drugs (DREADDs) to transiently inactivate projection neurons in the ACC or BLA in order to test the causal role of each area in confidence estimation or computation, and in learning under perceptual uncertainty.

We observe that rats are willing to tolerate longer delays to receive reward outcomes after faster and easier perceptual decisions involving more informative stimuli. Importantly, ACC is required for appropriate waiting according to the uncertainty of the visual stimulus and ensuing choice. Following ACC inhibition, post-decision waiting times are less sensitive to the strength of the visual evidence, and accuracy tracks the waiting times on a trial-by-trial basis less closely. In contrast, inhibition of BLA decreases rats' willingness to wait overall, regardless of the strength of the visual information and decision difficulty. Subsequent reversal learning is enhanced by confidence and both ACC and BLA inhibition block this enhancement. However, these effects happen through differential adjustments in response to reward feedback and in consistent use of learned rules on successive trials. Together, our results demonstrate dissociable contributions of ACC and BLA to computations of confidence and learning under uncertainty.

## Results

**Waiting time provides a proxy for confidence**. To assess confidence during perceptual choice with uncertain visual information, we used a novel experimental paradigm in which rats were first presented with a single Gabor patch with one of two possible dominant orientations (horizontal (H) and vertical (V)) embedded in noise (Fig. 1a, b). Perceptual uncertainty was manipulated using two parameters of the visual stimuli: (1) the signal-to-noise ratio (SNR), defined as the ratio of the contrast of the Gabor patch relative to the contrast of the added Gaussian noise; and (2) the overall contrast of both the Gabor patch and the added noise for a given SNR. These manipulations allowed us to modulate performance and confidence independently in order to design matched-performance different-confidence stimulus pairs used for learning[17–19]. Following stimulus presentation, rats reported the perceived orientation by nosepoking one of the two side compartments of the touchscreen based on a complementary stimulus–response rule (e.g., H → left and V → right). Following action selection, rats expressed confidence in their response via time wagering; that is, they could wait for a probabilistically delivered reward if confident or initiate a new trial otherwise (see Methods).

We found that during sessions with vehicle administration (see Viral constructs in Methods), our main control condition, waiting time and discrimination performance (the signal detection theoretic metric $d'$, a reliable measure of perceptual discrimination capacity; see Methods) were modulated by both the SNR and contrast of visual stimuli (Supplementary Fig. 1). More specifically, waiting time monotonically increased with SNR for any value of contrast (GLM contrast = 40: $p = 1.5 \times 10^{-57}$; $\beta_{SNR} = 1.13$, $p = 0.0002$; GLMcontrast = 60: $p = 8.7 \times 10^{-102}$; $\beta_{SNR} = 2.37$, $p = 7.7 \times 10^{-11}$; GLMcontrast = 80: $p = 7.3 \times 10^{-132}$; $\beta_{SNR} = 2.52$, $p = 2.7 \times 10^{-10}$) but this effect was modulated by overall stimulus contrast (GLM: $p = 1 \times 10^{-130}$; $\beta_{SNR \times contrast} = 0.03$, $p = 0.0008$). All our results hold when we perform our analyses for each contrast level separately (Supplementary Figs. 2 and 3).

Averaging over different contrast levels, we found that the probability of making a correct response was larger on average when SNR was larger (GLM: $p = 1.1 \times 10^{-51}$; $\beta_{SNR} = 0.054$, $p = 1.3 \times 10^{-29}$; $p$(correct) $= 0.69 \pm 0.06$, $0.74 \pm 0.05$, and $0.80 \pm 0.04$ for SNR of 2–4, respectively; where the value after $\pm$ denotes the standard deviation), and this also was true when considering individual contrast values separately (see Supplementary Note 1 and Supplementary Fig. 4). In addition, the number of re-initiations decreased as SNR increased (GLM: $p = 7.6 \times 10^{-11}$; $\beta_{SNR} = -0.01$, $p = 4.36 \times 10^{-6}$; fraction of re-initiated trials $= 0.17 \pm 0.02$, $0.16 \pm 0.03$, and $0.15 \pm 0.02$ for SNR of 2–4, respectively; Supplementary Note 2). Finally, the distribution of waiting time generally followed that of reward delivery (Supplementary Fig. 5). It is likely, however, that rats wait for a minimum amount of time (i.e., a threshold) before giving up or waiting longer. The distribution is skewed given the proximity of expected reward; possibly a temporal distortion as commonly reported in the interval timing literature[20] near the median time of reward dispensation, as shown in Supplementary Fig. 5. When we include all rats in our analysis, it appears most rats are waiting past this threshold (i.e., being more patient) as a common strategy (Supplementary Fig. 5b). Overall, these results illustrate that rats learned the task and used visual stimuli to make a choice and for post-choice wagering.

To show that waiting time to reinitiate a trial met several of the criteria for confidence readout, we analyzed this measure along with reaction time to make a response (i.e., the time between stimulus onset and nosepoke) during the control sessions. We found that waiting times were longer on trials with a larger SNR (GLM: $p = 2.5 \times 10^{-188}$; $\beta_{SNR} = 1.91$, $p = 1.2 \times 10^{-16}$; Fig. 2a). This effect has previously been reported for confidence reports in human and nonhuman primates[8,18,21], but not in rodents. In contrast, reaction time decreased with increased SNR (GLM:

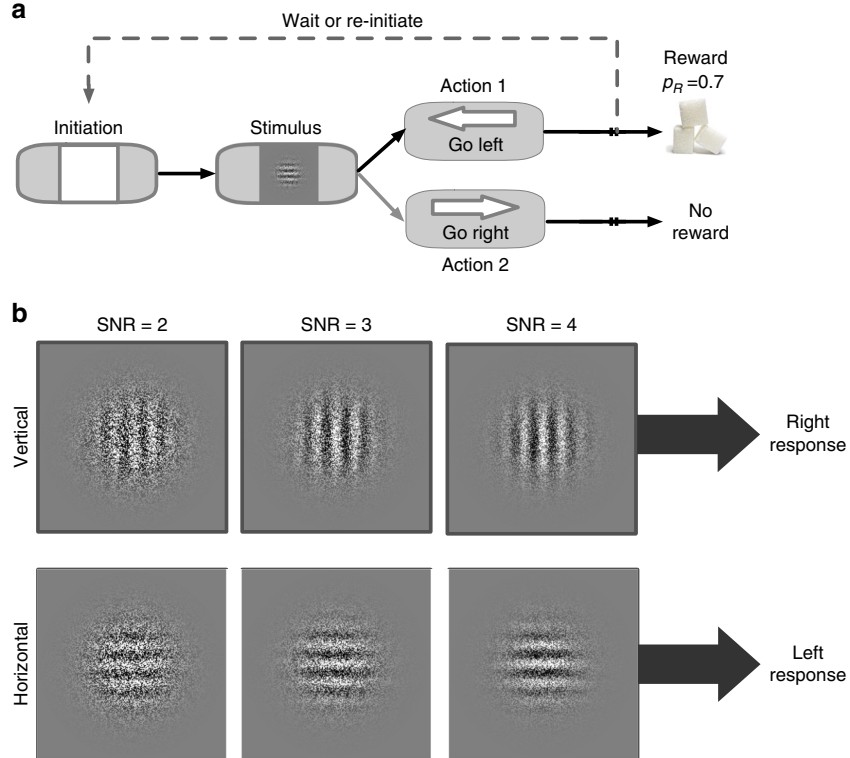

**Fig. 1** Schematic of the experimental paradigm, visual stimuli, and stimulus–response rule. **a** Timeline of each trial in the learning under perceptual uncertainty task. The rat first initiated a trial by nosepoking a white square in the center of the screen. The initiation stimulus then disappeared, and the rat was briefly presented (1 s) with a single horizontal (H) or vertical (V) Gabor patch masked by noise. Rats were required to report the dominant orientation (H or V) via nosepoke based on a complementary stimulus–response rule; e.g., H → left and V → right. Correct choices were rewarded probabilistically (on 70% of randomly selected trials), following variable delay times. After stimulus discrimination, rats could wait a self-timed delay in anticipation of reward or initiate a new trial. The initiation stimulus appeared on the touchscreen 2 s after a rat indicated its choice. **b** Examples of visual stimuli and one of the two stimulus–response rules. We refer to their discriminability as an SNR value reflecting the strength of visual signal (4, most discriminable; 3, moderately discriminable; 2, least discriminable). After discrimination of the visual stimulus, the rat makes a response (using the touchscreen) according to the rule H → Left and V → right

$p = 3.04 \times 10^{-216}$; $\beta_{SNR} = -0.32$, $p = 1.27 \times 10^{-78}$; Fig. 2b). In addition, waiting times were negatively correlated with reaction times on a trial-by-trial basis (GLM: $p = 4.98 \times 10^{-61}$; $\beta_{RT} = -7$, $p = 4.97 \times 10^{-61}$; Fig. 2c). Crucially, this negative correlation was significant for both correct and incorrect responses (GLM Correct: $\beta_{RT} = -8.66$, $p = 3.62 \times 10^{-51}$; GLM Incorrect: $\beta_{RT} = -4.16$, $p = 2.15 \times 10^{-25}$; Supplementary Fig. 6a, b). However, the correlation between waiting time and reaction time is associated more negatively for correct trials relative to incorrect trials (mean(diff) $= -0.074$ s; paired-sample $t$ test; $t(146) = -4.53$, $p = 1.2 \times 10^{-5}$; Supplementary Fig. 6c). A negative correlation between confidence and reaction time involving visual stimuli has also been reported in primates, but not in rodents[22–24].

To examine the relationship between accuracy and time wagering, we computed waiting time separately for correct and incorrect responses. We found that rats waited significantly longer following correct relative to incorrect responses, or trial type (diff (mean) = 4.74 s; GLM: $p = 1.31 \times 10^{-66}$; $\beta_{trial\ type} = 4.57$, $p = 2.46 \times 10^{-33}$; Fig. 2d). Consistent with this result, on trials with a re-initiation, rats discriminated more accurately when waiting times were longer for a given SNR (Pearson correlation; SNR = 2: $r = 0.017$, $p = 0.83$; SNR = 3, $r = 0.19$, $p = 0.017$; SNR = 4, $r = 0.19$, $p = 0.02$; Fig. 2a). In addition, waiting times on both correct and incorrect responses increased with larger SNR (GLM: Correct; $p = 1.95 \times 10^{-78}$; $\beta_{SNR} = 3.74$, $p = 1.40 \times 10^{-41}$; incorrect; $p = 1.77 \times 10^{-36}$; $\beta_{SNR} = 1.91$, $p = 2.33 \times 10^{-19}$). Finally, the normalized difference in waiting times between correct and incorrect

responses changed strongly (30–50%) for different SNR values (Fig. 2d inset). These results demonstrate that not only is waiting time sensitive to the strength of the visual information, but it also reflects rats' accuracy in discrimination.

Compatible with previous findings[25–27], we also found that reaction times decreased as SNR increased and were faster for correct responses relative to incorrect responses/discrimination (GLM: $p = 0.05$; $\beta_{SNR} = -0.069$, $p = 0.03$; Fig. 2e). The normalized difference in reaction times between correct and incorrect responses, however, changed only between 1 and 5% for different SNR values compared to 30–50% for waiting times (Fig. 2e inset). In addition, unlike waiting time, there was no significant correlation between performance and reaction time on a session-by-session basis (Pearson correlation; SNR = 2, $r = -0.1$, $p = 0.2$; SNR = 3, $r = 0.13$, $p = 0.1$; SNR = 4, $r = 0.01$, $p = 0.82$; Fig. 2b). Importantly, we found similar results when we performed all above analyses for each value of contrast separately (Supplementary Fig. 2). Moreover, independently of SNR and contrast levels, waiting time increased with discrimination performance (Supplementary Fig. 1b). Finally, all these results hold when we used normalized waiting time and reaction time (see Supplementary Note 3).

Together, our results illustrate that waiting time reflects confidence in a perceptual discrimination with much higher fidelity than that of reaction time, to include the proportional nature of confidence and accuracy. Our findings thus extend previous observations in primates[28] to rodents, and suggest that

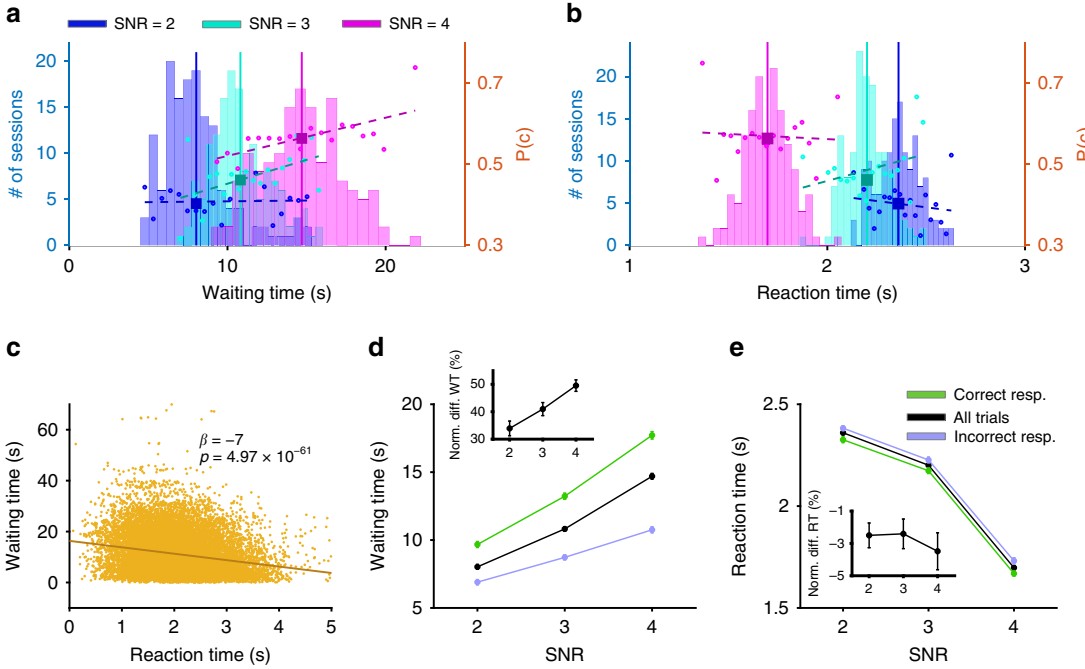

**Fig. 2** Waiting time serves as a proxy for confidence that is more sensitive than reaction time. **a** Waiting time before reinitiation increases as SNR increases. Plotted are the distributions of waiting times for each SNR for all sessions, and probability correct ($P(c)$) as a second axis: 2 (blue), 3 (cyan), and 4 (magenta), following vehicle administration. Solid lines show the median of each distribution. **b** Reaction time decreases as SNR increases. Plotted are the distributions of reaction time for all sessions, and probability correct ($P(c)$) as a second axis. Conventions are the same as in panel **a**. **c** Waiting time before reinitiation of a new trial is negatively correlated with reaction time to make a choice. Waiting time is plotted as a function of the reaction time for all trials and all rats. Each data point is a trial in a session following vehicle administration. **d** Waiting time is larger for correct compared to incorrect responses for any SNR. Plotted is waiting time for all trials (black), correct trials (green), and incorrect trials (purple) for different SNR. The inset shows the relative difference in waiting time between correct and incorrect responses for different SNR. Error bars show the S.E.M. over sessions (typically smaller than the symbol). **e** Reaction time only weakly reflects the accuracy of the response. Plotted is the reaction time for all trials and separately for correct and incorrect responses. Conventions are the same as in panel **d**. The inset shows the relative difference in reaction time between correct and incorrect responses for different SNR. Overall, response accuracy is reflected in waiting time an order of magnitude better than in reaction time. Source data are provided as a Source Data file

waiting time in our paradigm can also serve as a proxy for decision confidence[11].

**Dissociable contributions of BLA and ACC to time wagering.** We expressed Gi-coupled DREADDs on projection neurons of ACC and BLA (Fig. 3). After allowing time for transduction, we injected rats with clozapine-N-oxide (CNO) prior to a subset of testing sessions to inhibit these brain regions, using a within-subject design. In addition, to confirm the effect of CNO using ex vivo electrophysiological recording, we prepared a separate group of rats ($n = 3$) with ACC DREADDs using identical procedures. We found a significant reduction in field potential after CNO application only in the transfected slices (Supplementary Fig. 7).

For rats performing the main experiments, we observed a significant interaction of drug condition (vehicle and CNO), targeted brain region (BLA or ACC), and SNR on waiting time when combining correct and incorrect responses (GLM: $p = 10^{-17}$; $\beta_{\text{drug×region×ratio}} = 1.84$, $p = 7.4 \times 10^{-6}$). We found no significant effect of the targeted brain region (BLA vs. ACC) in sessions following vehicle administration on waiting time (GLM: $p = 2.5 \times 10^{-188}$; $\beta_{\text{region}} = -0.93$, $p = 0.335$). In contrast, following CNO administration, we observed a significant effect of the targeted brain region on waiting time (GLM: $p = 6.36 \times 10^{-305}$; $\beta_{\text{region}} = 1.66$, $p = 5.11 \times 10^{-55}$; Fig. 4a, b).

An analysis of waiting time for different SNR values averaged across all trial types (correct and incorrect) revealed a significant drug × SNR × brain region interaction (GLM: $p = 7.07 \times 10^{-282}$; $\beta_{\text{drug×SNR×region}} = 1.66$, $p = 1.94 \times 10^{-5}$). When the trial type

(correct vs. incorrect) was included as a within-subject factor, there was similarly a significant trial type × drug × brain region interaction (GLM: $p = 10^{-17}$, $\beta_{\text{trialtype×drug×region}} = 1.84$, $p = 7.40 \times 10^{-6}$). These results show that inhibition of ACC or BLA affect rats' willingness to wait depending on SNR (perceptual uncertainty) as well as based on whether their response was correct or incorrect.

Given these interaction effects, we then measured the influence of inhibition of ACC and BLA on waiting time separately for each value of SNR. We found that overall, inhibition of ACC significantly increased waiting time compared to vehicle (diff(mean) = 3.45 s; GLM: $p = 4.98 \times 10^{-43}$; $\beta_{\text{drug}} = 3.45$, $p = 4.98 \times 10^{-43}$; Fig. 4a, c). In contrast, inhibition of BLA significantly reduced rats' overall willingness to wait before reinitiation of a new trial (diff(mean) = $-4.59$ s; GLM: $p = 4.87 \times 10^{-76}$; $\beta_{\text{drug}} = -4.58$, $p = 4.86 \times 10^{-76}$; Fig. 4b, c). Importantly, we found a significant interaction of SNR by drug condition (CNO vs. vehicle administration) for ACC (GLM: $\beta_{\text{SNR×drug}} = -2.19$, $p = 1.78 \times 10^{-10}$), indicating that the observed increase in waiting time due to ACC inhibition depended on SNR. In contrast, the decrease in waiting time due to BLA inhibition was not SNR-specific (GLM: $\beta_{\text{SNR×drug}} = -0.3434$, $p = 0.16$), indicating that BLA inhibition increased sensitivity to delays (or equivalently increased impulsivity), as has been reported before[29]. Finally, we found similar results when we performed our analyses for each value of contrast separately (Supplementary Fig. 3a–c).

Despite strong, dissociable effects on waiting time, inhibition of ACC and BLA did not change the overall task performance,

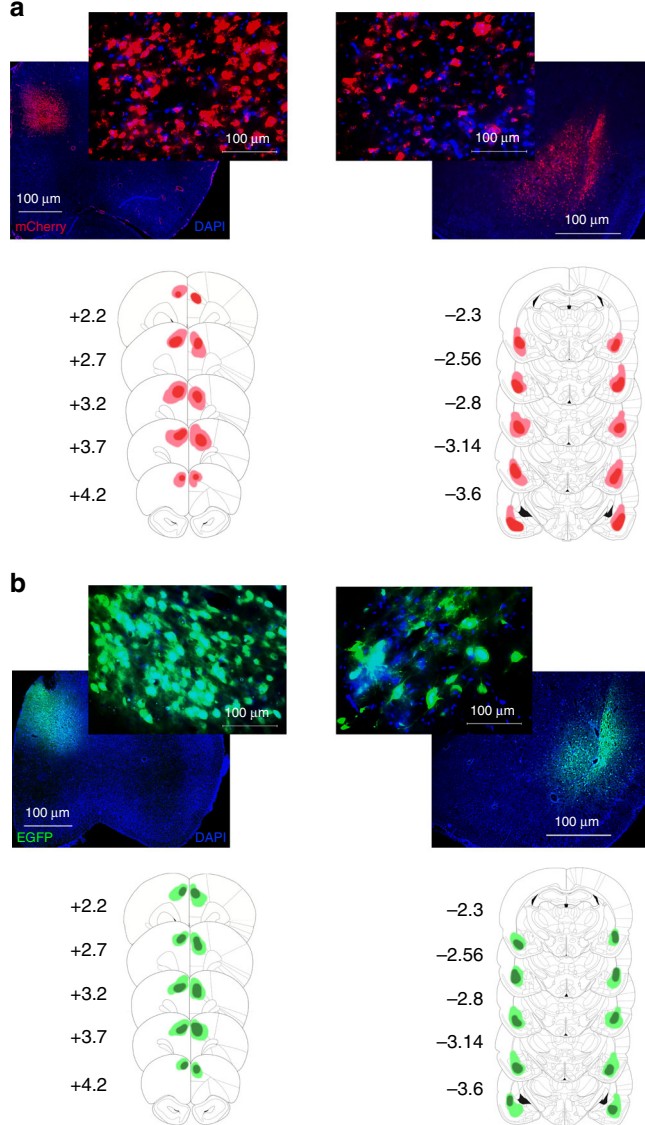

**Fig. 3** Expression of inhibitory DREADDs and null virus EGFP in anterior cingulate cortex and basolateral amygdala. **a** Top, representative expression of hM4Di-mCherry DREADDs under CaMKIIa is shown for ACC (left) and BLA (right). Bottom, reconstructions of maximum viral spread for all rats. Numerals depict + anterior–posterior (AP) level relative to Bregma. Scale bars 1000 with 100 μm (inset). **b** Top, representative expression of eGFP (null virus) under CaMKIIa is shown for ACC (left) and BLA (right). Bottom, reconstructions of maximum viral spread for all rats. Numerals depict + anterior–posterior (AP) level relative to Bregma. Scale bars 1000 with 100 μm (inset). Source data are provided as a Source Data file

discrimination accuracy, response bias, or reaction time. First, probability of correct response was not significantly different between vehicle and CNO administration (GLM: $p = 0.04$; $\beta_{drug} = 0.003$, $p = 0.57$; Fig. 5a). Second, we computed discrimination performance, or $d'$, and found this measure was also not significantly different between vehicle and CNO administration (GLM: $p = 4.19 \times 10^{-24}$; $\beta_{drug} = -0.26$, $p = 0.21$; Fig. 5b). Importantly, we observed a strong and significant correlation between discrimination accuracy $d'$ following CNO administration and following vehicle administration in rats with DREADDs expressed either in ACC (Pearson correlation; $r = 0.847$, $p = 1.25 \times 10^{-6}$) or BLA (Pearson correlation; $r = 0.767$, $p = 1.21 \times 10^{-5}$). Moreover, contrary to the significant effect of SNR on

discrimination accuracy $d'$ (GLM: $p = 4.19 \times 10^{-24}$; $\beta_{ratio} = 0.38$, $p = 2.38 \times 10^{-11}$), we found no significant main effect or interaction of drug condition (CNO vs. vehicle administration) and targeted brain region (ACC vs. BLA) on $d'$ (GLM: $\beta_{drug} = -0.26$, $p = 0.21$; $\beta_{region} = 0.04$, $p = 0.84$), indicating that perceptual discrimination was not affected by ACC or BLA inhibition. Third, we found no significant effect of drug condition (CNO vs. vehicle) and targeted brain region on the decision criterion (i.e., the response bias[30]; GLM: $p = 0.19$; $\beta_{drug} = -0.03$, $p = 0.23$; $\beta_{region} = -0.007$, $p = 0.74$). Finally, ACC and BLA inhibition failed to affect task engagement and perceptual processing speed as evidenced by the lack of change in the distributions of reaction time (GLM: $p = 10^{-16}$; $\beta_{drug} = -0.01$, $p = 0.81$; $\beta_{region} = 0.001$, $p = 0.98$; Fig. 4f), and these responses also did not differ by trial type (correct vs. incorrect; Correct: GLM: $p = 2.01 \times 10^{-208}$; $\beta_{ratio} = -0.33$, $p = 1.01 \times 10^{-81}$; Incorrect: GLM: $p = 3.44 \times 10^{-196}$; $\beta_{ratio} = -0.32$, $p = 3.14 \times 10^{-72}$; Fig. 4g, h). This pattern also held for each value of contrast separately (Supplementary Fig. 3d–f). Together these findings demonstrate that the observed effects of ACC and BLA inhibition on waiting times were not attributable to changes in decision-making processes related to visual discrimination.

Finally, in additional control conditions, we also tested whether the presence of active virus was essential for the observed changes, and whether vehicle administration alone could cause changes in behavior. To do so, we measured behavioral responses in the rats with expressed DREADDs but without the administration of vehicle (no-injection control prior to reversal) and in rats with null virus and compared them with those under vehicle administration (i.e., the main control condition). We found that waiting time and reaction time did not differ between vehicle administration and no-injection control (Supplementary Note 4 and Supplementary Fig. 8). In addition, we found that the observed effects of CNO depended on the presence of active virus (Supplementary Note 5 and Supplementary Fig. 9).

Together, these results suggest that ACC contributes to the computations and transmission of confidence to influence post-decision behavior. In contrast, BLA mainly increases waiting time independently of perceptual uncertainty perhaps by controlling impulsive behavior during choice under uncertainty.

**ACC-specific role in evaluation for confidence report.** Although BLA inhibition decreased waiting time, this measure still increased with greater SNR (GLM: $p = 2.32 \times 10^{-113}$; $\beta_{SNR} = 1.59$, $p = 5.44 \times 10^{-23}$; Fig. 4c) similar to the behavioral pattern observed under vehicle administration (GLM: $p = 1.63 \times 10^{-189}$; $\beta_{SNR} = 1.92$, $p = 3.16 \times 10^{-32}$). This suggests that BLA contributes to shifting the influence of confidence on post-decision processes making the animal more patient irrespective of their confidence. In contrast, ACC inhibition rendered waiting time mainly insensitive to SNR (GLM: $p = 7.17 \times 10^{-30}$; $\beta_{SNR} = -0.27$, $p = 0.22$; Fig. 4c), with a significant effect of SNR on correct trials (GLM: $p = 6.09 \times 10^{-7}$; $\beta_{ratio} = 1.22$, $p = 6.09 \times 10^{-7}$; Fig. 4d), but not for incorrect trials (GLM: $p = 0.2$; $\beta_{ratio} = -0.27$, $p = 0.2$; Fig. 4e). Critically, ACC inhibition did not cause a uniform increase in waiting time for all SNR; the $z$-scored waiting time for SNR = 4 in ACC inhibition is even decreased compared to vehicle (two-sided $t$ test: $t(213) = 12.41$, $p = 5.4 \times 10^{-27}$). This suggests that ACC inhibition removes the sensitivity to SNR and does not simply increase waiting time. In addition, the distributions of waiting times were quite symmetric in all conditions, and their means and medians were very far from the 40 s maximum waiting time (Supplementary Fig. 5). This provides evidence that the observed insensitivity of waiting time to SNR after ACC inhibition was unlikely due to a ceiling effect. Interestingly, ACC inhibition

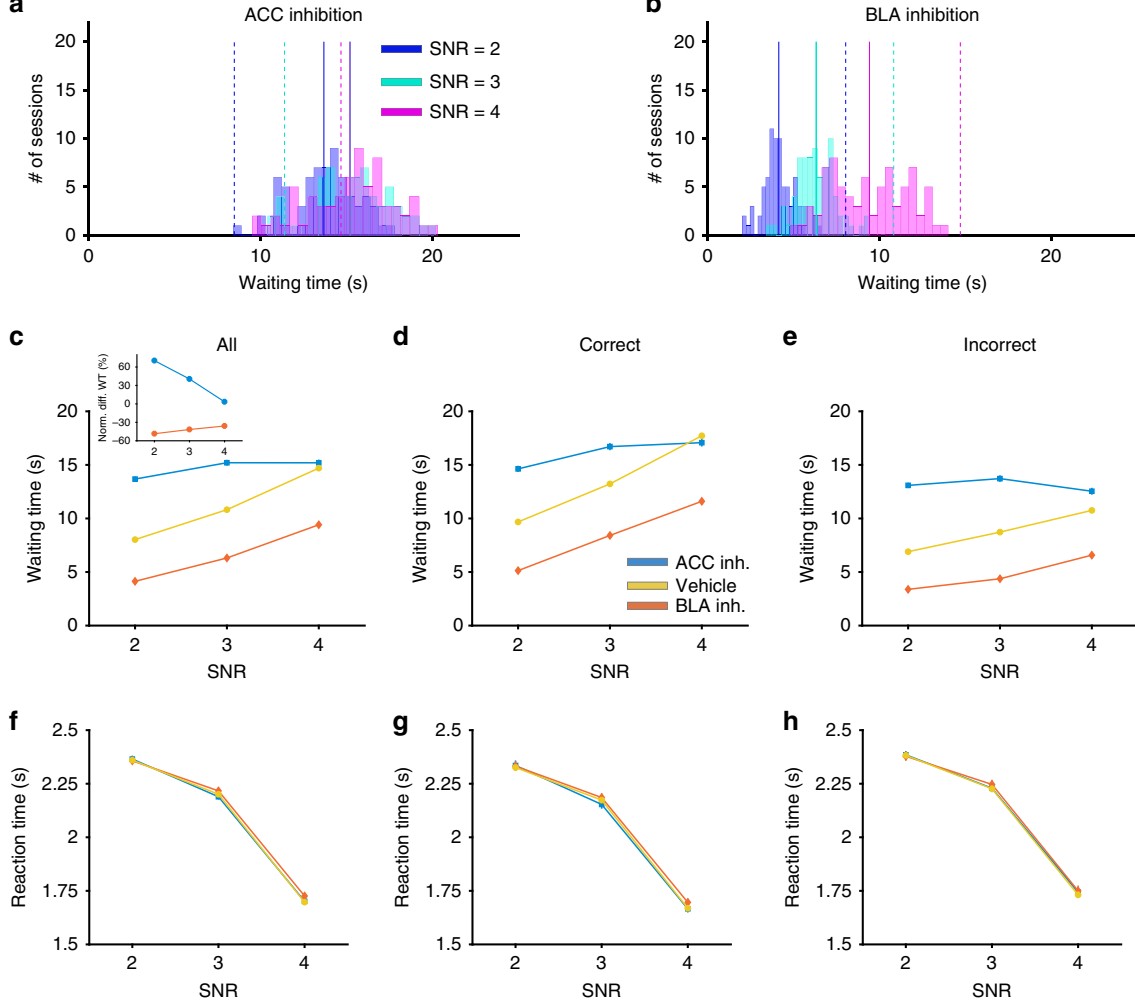

**Fig. 4** ACC and BLA inhibition result in opposing changes in waiting time, but no change in reaction times. **a** Waiting time increases following ACC inhibition. Plotted are the distributions of the waiting time separately for each SNR, following ACC inhibition. Solid lines show the median of each distribution. Dashed lines show the median of the same condition but following vehicle administration as is shown in Fig. 2a. **b** Waiting time decreases following BLA inhibition. Same as in panel A but for sessions following inhibition of BLA. **c** Inhibition of ACC renders waiting times insensitive to the strength of sensory signal, whereas BLA inhibition shift waiting times. Plotted is the waiting time for all trials as a function of SNR following vehicle administration (yellow), inhibition of BLA (orange), and inhibition of ACC (blue). **d**, **e** Same as in panel (**c**) but only on trials in which a correct (**d**) or incorrect (**e**) response was made. Error bars show the S.E.M. over sessions (typically smaller than the symbols). **f** Reaction time is unaffected by inhibition of either ACC or BLA. Plotted is the reaction time for all trials as a function of SNR following vehicle administration (yellow), inhibition of BLA (orange), and inhibition of ACC (blue). **g**, **h** Same as in panel (**f**) but only on trials in which a correct (**g**) or incorrect (**h**) response was made. Source data are provided as a Source Data file

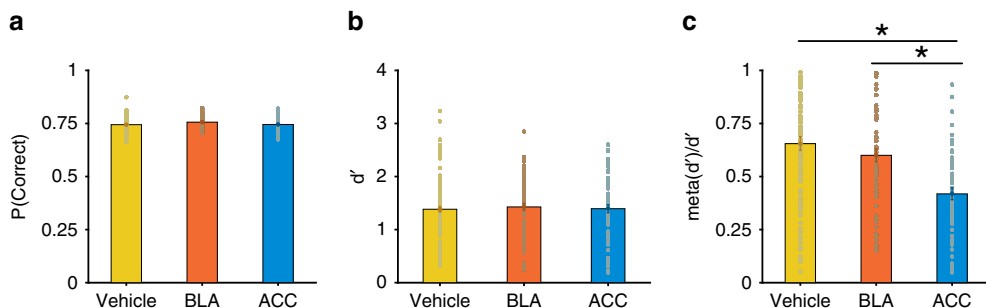

**Fig. 5** Discrimination performance is intact following inhibition of either ACC or BLA, but metacognitive efficiency is decreased following inhibition of ACC only. **a** Plotted is the probability of correct responses for the sessions following vehicle administration (yellow), sessions following BLA inhibition (orange), and sessions following inhibition of ACC (blue). Error bars show S.E.M. **b** Plotted is discrimination performance, *d′* in the three conditions defined in panel (**a**). **c** Plotted is the metacognitive efficiency (meta—*d′*/*d′*) in the three conditions defined in panel (**a**). (*) indicates $p < 0.05$ in one-way ANOVA test. Source data are provided as a Source Data file

renders waiting time even insensitive to the contrast level of visual stimuli such that for higher contrast and higher SNR, waiting time following inhibition dropped below the control condition (Supplementary Fig. 3c). These findings demonstrate that ACC is involved in modifying visual uncertainty, perhaps via gain modulation, in order to compute perceptual uncertainty and to influence post-decision processes based on the latter.

To further test this, we computed metacognitive efficiency (meta—$d'/d'$), that assesses how well waiting time tracks discrimination performance ($d'$) across trials[31], or equivalently, the trial-by-trial correspondence of accuracy and waiting time (see Methods). A one-way ANOVA resulted in a significant difference between groups (one-way ANOVA: $F(2,267) = 12.65$, $p = 5.6142 \times 10^{-6}$). Following BLA inhibition, meta—$d'/d'$ was not significantly different from vehicle (one-way ANOVA: diff (mean) $= 0.0509$, $p = 0.4430$). However, following ACC inhibition, meta—$d'/d'$, was significantly lower than vehicle (one-way ANOVA: diff(mean) $= -0.2366$, $p = 1.6438 \times 10^{-6}$; Fig. 5c). These results demonstrate that confidence report following ACC inhibition becomes less sensitive to the accuracy of the trial and thus further suggest that ACC is involved in the computation of confidence. Consistent with these results, we found that the trial-by-trial correlation between waiting time and reaction time was weaker following ACC inhibition compared to BLA inhibition (Supplementary Fig. 10 and Supplementary Note 6).

Collectively, these results suggest that whereas inhibition of BLA decreases waiting time, this effect is most likely due to a general delay aversion or an increase in impulsive choice, because rats are still able to appropriately scale their waiting times according to performance and trial difficulty. In contrast, inhibition of the ACC renders rats' waiting times relatively insensitive to discrimination accuracy ($d'$) and SNR, suggesting that this region meaningfully participates in estimating the reliability of visual stimuli and consequently, computing and reporting confidence. Taken together with the results we provide in the previous section, we show that here in rats we are able to interfere with and dissociate first order (discrimination performance) from second order (metacognition) processes, as has been done in nonhuman and human primates[18,19,32,33].

**Confidence enhances reversal learning.** For the reversal learning phase, we found two pairs of SNR and contrast level in which a given rat demonstrated equivalent accuracy (i.e., matched discrimination accuracy, or $d'$). Critically, although performance was equivalent for these two pairs, the rat on average waited longer for one of the pairs. We then randomly chose one of these pairs for each rat for the reversal learning phase. If the rat waited on average less time for one pair over the other, this was designated the low-confidence (LC) condition. If the rat waited on average more time than the other, this was designated the high-confidence (HC) condition. We also confirmed that rats in the HC and LC conditions received the same amount of reward in the session prior to reversal. During reversal learning, the stimulus that signaled the possibility for a trial reinitiation (i.e., a white square in the middle panel) was removed. This change was introduced to ensure that the feedback rats received in the reversal learning phase was not a function of waiting time but rather a function of accuracy in learning the reversal policy. Correct responses, now under a reversed stimulus–response rule, were reinforced probabilistically as before (70% of the time). We calculated $d'$ and confidence using only the data from sessions that were not preceded by injections (i.e., no-injection control prior to reversal) in order to assign rats to HC and LC conditions.

We performed several analyses to ensure that the only difference between HC and LC conditions was the confidence reported via waiting time. First, we found that $d'$ for HC and LC conditions were not significantly different for each of the stimuli that was administered after reversal, i.e., the contrast-SNR pairs that were chosen for reversal were not associated with different $d'$ before reversal (Stepwise GLM: $p = 0.11$; $\beta_{confidence} = 0.43$, $p = 0.32$; Supplementary Fig. 11a). Second, $d'$ for HC and LC conditions were not significantly different across all contrast-SNR pairs (GLM: $p = 8.2 \times 10^{-16}$; $\beta_{confidence} = -0.14$, $p = 0.92$; Supplementary Fig. 11b). Third, metacognitive efficiency (meta—$d'/d'$) across HC and LC conditions was not significantly different for the specific contrast-SNR pair that was used after reversal (Stepwise GLM: $p = 0.44$; $\beta_{confidence} = -0.1$, $p = 0.44$; Supplementary Fig. 11c). Fourth, rats in both HC and LC conditions acquired equal amount of reward in the no-injection control session for the specific pair of contrast and SNR values (Stepwise GLM: $p = 0.01$; $\beta_{confidence} = -0.2$, $p = 0.08$; Supplementary Fig. 11d). Finally, we found that HC and LC conditions were different in waiting time, reflecting confidence, for the specific contrast-SNR pair that was used after reversal (Stepwise GLM: $p = 2.7 \times 10^{-31}$; $\beta_{confidence} = -28.3$, $p = 2.81 \times 10^{-9}$; Supplementary Fig. 11e). Together, these results illustrate that the only difference between HC and LC conditions was the confidence.

To assess the effect of perceptual uncertainty on learning, we analyzed the probability of correct response for both HC and LC conditions for rats receiving vehicle or CNO injections, and for ACC or BLA as the targeted brain region. We observed significant interactions of drug by confidence level (GLM: $p = 2.95 \times 10^{82}$; $\beta_{drug \times confidence} = -0.22$, $p = 3.71 \times 10^{-7}$ as well as drug by trial bin ($\beta = -0.009$, $p = 0.0001$). There was also a significant main effect of trial bin ($\beta = 0.036$, $p = 1.13 \times 10^{-44}$), illustrating that all rats were able to learn the new stimulus–response rule. Similarly, there was a significant main effect of confidence on learning ($\beta = 0.15$, $p = 3.4 \times 10^{-6}$).

To identify how learning and choice strategies were affected by confidence, we first compared learning between HC and LC conditions following vehicle administration. We found that rats in the HC condition performed better than the rats in the LC condition following vehicle administration (diff(mean) $= 0.1063$; GLM: $p = 1.13 \times 10^{-39}$; confidence: $\beta = 0.151$, $p = 2.71 \times 10^{-6}$). This improvement in performance was due to faster learning in the HC compared to LC condition (chi-square test of ratio; $p = 5.87 \times 10^{-26}$; Fig. 6a inset) and the steady state of performance was not affected by confidence (chi-square test of ratio; $p = 0.27$).

The observed faster learning occurred simultaneously with an increase in selection of the correct stimulus–response rule following selection of this rule and being rewarded on the preceding trial (Win–Stay; Permutation test; $p = 9.7 \times 10^{-10}$; Fig. 6b). In addition, animals increased their tendency to switch from the incorrect to correct stimulus–response rule following unrewarded trials when the response on the preceding trial was incorrect (Lose–Switch after incorrect; Permutation test; $p = 0.046$). The improvement in learning due to higher confidence was also accompanied by a decrease in switch from the correct to incorrect stimulus–response rule when the response on the preceding trial was correct but not rewarded (Lose–Switch after correct; note that 30% of correct responses were not rewarded by design; Permutation test; $p = 3.3 \times 10^{-4}$). We also compared the tendency of the animals to repeat the same stimulus–response rule as in the previous trial beyond what is expected by chance, measured by the rule-based repetition index (RRI[34]; see Methods). We found that RRI was larger for the HC relative to LC condition (Permutation test; $p = 0.029$), indicating that animals were more consistent/persistent in their behavior

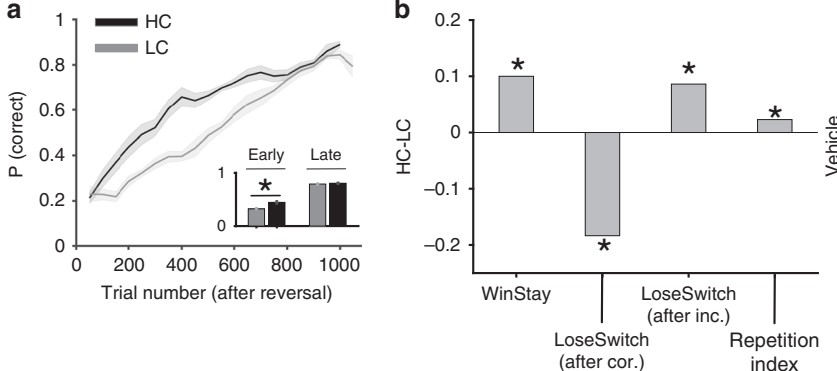

**Fig. 6** Influence of decision confidence on subsequent reversal learning. **a** Learning curves showing performance (probability of correct response) following reversal for rats prepared with DREADDs after vehicle administration. Plots shows the performance across all rats in high-confidence (HC; black) and low-confidence (LC; gray) conditions averaged over a sliding window of 100 trials. The inset shows the average performance in early (first half of the trials) and late (last quarter of the trials) trials, demonstrating that higher confidence improved the rate of learning but did not change the steady state. (*) indicates a significant difference in median between the two conditions (chi-square test of ratio, $p < 0.05$). Error bars show the S.E.M. over sessions. **b** The influence of confidence on learning strategies and perseveration. Plotted are the difference in the proportions of Win–Stay, Lose–Switch following correct but unrewarded responses, Lose–Switch after incorrect responses, and rule-based repetition index between HC and LC conditions in the first half of trials after the reversal. Source data are provided as a Source Data file

(following a specific rule) under higher confidence. Together, these results suggest that confidence can improve learning strategies from all possible outcomes and moreover, can increase consistency in following learned stimulus–response rules. To our knowledge, the observed enhancing effect of perceptual confidence on learning has been reported in humans[3] but not in rodents, and the effects on rule consistency are novel.

**Both BLA and ACC support reversal learning**. We next compared overall learning across different DREADDS inhibition conditions. In contrast to the control conditions, the overall performance over time in the LC condition was significantly better than in the HC condition following CNO treatment (diff(mean) = 0.0456; GLM: $p = 5.62 \times 10^{-46}$; $\beta_{\text{confidence}} = -0.0719$, $p = 0.01$). Furthermore, the reduction in performance was not significantly different between ACC and BLA as the targeted region (GLM: $p = 5.78 \times 10^{-42}$; $\beta_{\text{region}} = -0.035$, $p = 0.396$; Fig. 7a, c). To estimate the rate of learning, we fit rats' learning curves after the reversal using an exponential function (see Methods for details). We observed that the exponent or the learning parameter (α; which reflects the rate of learning) was significantly different between CNO and vehicle administration sessions in the HC condition (confidence interval; ACC; CNO, [15.2,15.78], vehicle, [9.42,10.04]; BLA; CNO, [13.28,16.20], vehicle, [8.23,9.23]). However, there was no significant difference in the learning parameter between sessions following CNO and vehicle administration in the LC condition (confidence interval; ACC; CNO, [12.52,13.75], vehicle, [12.36,15.18]; BLA; CNO, [12.78,14.12], vehicle, [12.63,14.33]). Therefore, following inhibition of either ACC or BLA, the rate of learning decreased in the HC but not the LC condition. Together, these results indicate that whereas rats could still learn a new stimulus–response rule after the inhibition of the ACC or BLA, these brain regions contribute to an enhancing effect (i.e., the use) of perceptual confidence on learning despite matched $d'$ across HC and LC conditions.

**BLA and ACC use confidence for learning strategy**. Results reported above suggest that both types of DREADDs inhibition (in ACC or BLA) removed the benefit of confidence on learning. To better reveal similarities and differences between the effects of ACC and BLA inhibition on learning and their dependence on

confidence, we computed changes in the learning strategies and consistency in following stimulus–response rules after ACC and BLA inhibition, separately for rats in the LC and HC conditions.

First, we examined the effect of positive feedback on the tendency to stay on the correct stimulus–response rule (Win–Stay). We found that inhibition of either ACC or BLA resulted in a reduction in the difference in Win–Stay between HC and LC conditions (Permutation test; ACC: $p = 5.43 \times 10^{-4}$; BLA: $p = 5.83 \times 10^{-12}$; Fig. 7b, d) but this reduction was stronger for BLA inhibition (Permutation test; $p = 0.0048$). This suggests that both ACC and BLA inhibition attenuate reversal learning by reducing the tendency to repeat a rewarded stimulus–response rule due to higher confidence. Therefore, although both ACC and BLA contribute to mediating the effect of confidence on learning from positive feedback, the stronger attenuation and reversal of this effect following BLA inhibition illustrates a more prominent role for BLA in learning under uncertainty.

Secondly, we analyzed the effect of negative feedback on switching from the previous stimulus–response rule, separately for when the previous rule was correct (Lose–Switch after correct) and when the previous rule was incorrect (Lose–Switch after incorrect). We examined these two types of trials separately because as we showed above, confidence has differential effects on these trials (Fig. 6b). We observed that both ACC and BLA inhibition reversed the effect of confidence (i.e., difference between HC and LC conditions) on learning from negative feedback; however, this was only significant for BLA (Permutation test; ACC Lose–Switch after correct: $p = 0.23$; ACC Lose–Switch after incorrect: $p = 0.44$; BLA Lose–Switch after correct: $p = 7.4 \times 10^{-4}$; BLA Lose–Switch after incorrect: $p = 0.0044$; Fig. 7b, d). In addition, the effect of confidence on Lose–Switch after an incorrect response became more negative after ACC inhibition compared to BLA inhibition (Permutation test; $p = 0.0051$), indicating a stronger role for ACC in learning from negative feedback when the response was incorrect. Together, these results suggest that BLA has a more pronounced role in mediating the effect of confidence in learning from positive feedback whereas ACC is more involved in mediating the effect of confidence in learning from negative feedback.

Finally, we evaluated the consistency in using learned stimulus–response rules using the RRI. We found that BLA but not ACC inhibition reversed the effect of confidence on RRI

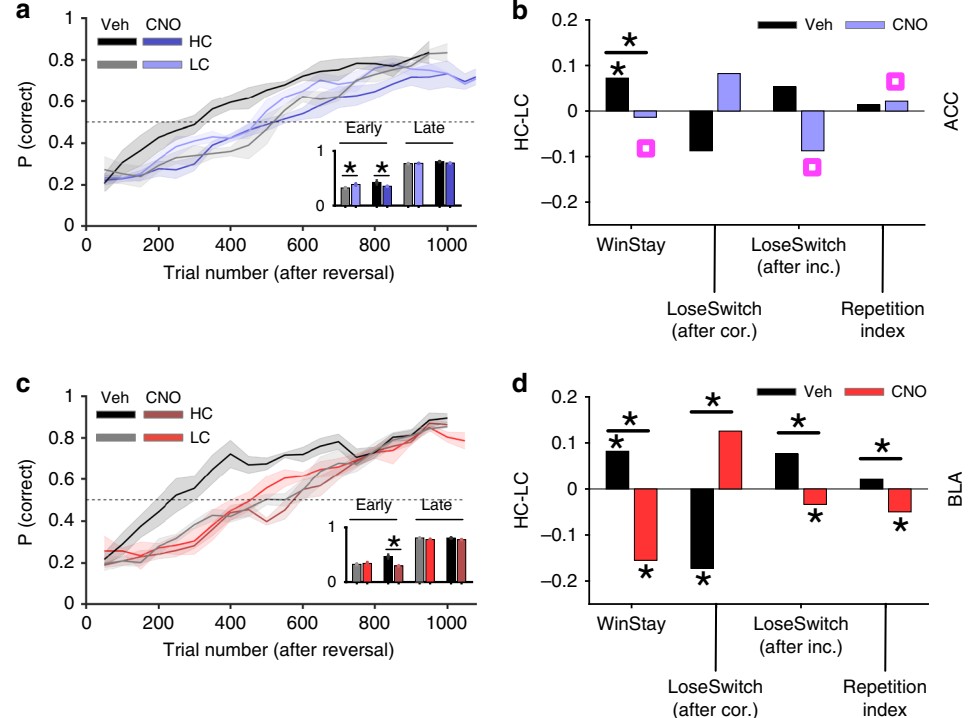

**Fig. 7** ACC and BLA inhibition differentially modulate the effects of perceptual uncertainty on learning strategies and perseveration. **a** Learning curves (probability of correct response) after a reversal for rats prepared with ACC DREADDs following CNO (blue) or vehicle (Veh; black) administration. Plot shows the performance across all rats averaged over a sliding window of 100 trials for high-confidence (HC) and low-confidence (LC) conditions. The inset shows the average performance in early (first half of the trials) and late (last quarter of the trials) trials, demonstrating that either perceptual uncertainty or ACC inhibition decrease the rate of learning. Following ACC inhibition, rats eventually reach a similar performance level compared to the control condition (vehicle administration). (*) indicates a significant difference in median between the two conditions (chi-square test of ratio, $p < 0.05$). Error bars show the S.E.M. over sessions. **b** Win–Stay, Lose–Switch following correct response, Lose–Switch after incorrect response, and rule-based repetition index for the HC and LC conditions in ACC DREADDs during the first half of trials after the reversal. ACC inhibition only removes the benefit of confidence on Win–Stay but weakens the effect of confidence on learning from negative feedback or consistency in rule selection. (*) indicates a median significantly different from zero or a significant difference in median between the two conditions (Permutation test, Bonferroni corrected, $p < 0.01$). Magenta squares indicate a significance difference between ACC and BLA inhibition (Permutation test, $p < 0.05$). **c** Learning curves (probability of correct response) after a reversal for rats prepared with BLA DREADDs following CNO (red) or vehicle (black) administration. BLA inhibition decreases the rate of learning but eventually rats reach a similar performance level compared to the vehicle administration condition. Error bars show the S.E.M. over sessions. **d** The same as in panel B but for BLA DREADDs. Unlike ACC inhibition, BLA inhibition reverses the benefits of confidence on all learning strategies and consistency in rule selection. Source data are provided as a Source Data file

(Permutation test; ACC: $p = 0.36$; BLA: $p = 4.6 \times 10^{-4}$; Fig. 7b, d), and this effect was more negative after BLA than ACC inhibition (Permutation test; $p = 4.4 \times 10^{-4}$). This indicates that BLA, but not ACC, is important for mediating the effect of confidence in consistently using learned stimulus–response rules under perceptual uncertainty. Importantly, there was no significant difference between the effects of confidence on all aforementioned strategies in sessions following vehicle administration based on the targeted region (Permutation test; Win-Stay: $p = 0.31$; Lose-Switch after correct: $p = 0.10$; Lose-Switch after incorrect: $p = 0.35$; RI: $p = 0.36$; Fig. 7b, d).

Together, these results reveal dissociable effects of ACC and BLA inhibition on learning under uncertainty. Importantly, only BLA inhibition consistently reversed the benefit of confidence on learning from both positive and negative feedback. This suggests that BLA is directly involved in confidence-dependent learning (and not estimation) because BLA inhibition globally shifts confidence readout, as shown earlier. Different from BLA effects, ACC has a more specific role in supporting learning, mainly from positive feedback (Win-Stay), following a correct response and to lesser extent from negative feedback perhaps by making confidence computation sensitive to the level of perceptual uncertainty, as suggested by our results on confidence readout. In

addition, the consistency in using learned stimulus–response rules did not differ by confidence condition following ACC inhibition, whereas BLA inhibition made rats less likely to apply the learned rules under higher confidence.

## Discussion

We examined the causal roles of ACC and BLA in confidence report and learning under perceptual uncertainty. We trained rats to report the orientation of ambiguous visual stimuli based on a learned stimulus–response rule and we read out their confidence in choice using a time wagering task. Despite previous suggestions for possible roles of confidence in learning[3,28], such roles had not been explored directly in any animal model. We found that high confidence in a perceptual decision can boost subsequent reversal learning of stimulus–response rules using reward feedback, even when we controlled for signal processing capacity (i.e., task performance). Critically, all rats were able to learn new reward contingencies upon the change in stimulus–response mapping, but the learning was faster in the group of rats that exhibited higher confidence at the onset of reversal. We found that the BLA and ACC are both required for the enhancement of learning by perceptual certainty or confidence.

Several studies find that confidence increases as stimulus strength increases for correct trials but decreases as stimulus strength increases for error trials. This opposing relation on correct and error trials has been deemed a statistical signature (also referred to as divergence signature) of confidence report[35]. We did not observe this pattern in our data and instead found that waiting time (as a proxy for confidence) increased as stimulus strength (i.e., SNR) increased for both correct and error trials. However, there are recent experimental and theoretical studies showing that confidence can also increase on error trials with larger sensory evidence and thus, suggest that the opposing pattern on correct and error is not necessarily the defining signature of confidence as previously assumed[23,32,36,37]. More specifically, there are recent studies based on Bayesian framework that argue this divergence signature of confidence disappears under different conditions of stimuli, noise, and task structure[38,39]. Relevant to the present results, a study that similarly observed a violation of this divergence featured a stimulus composed of a feature that was used to probe accuracy and an orthogonal manipulation of stimulus strength[32,39]. In fact, this is the case in our experiment in which we assessed accuracy in perceptual judgment by changing SNR and contrast of the stimuli as two orthogonal manipulations. In addition, bounded accumulation models have been used to explain how the time to decision influences confidence and why confidence in error trials increases with stronger stimulus strength[37,38]. Taken together, observing a violation of this divergence in our experiment is a feasible outcome based on the aforementioned studies.

In rats, perceptual metacognition has been previously assessed within olfactory and auditory[2,10,11] but not visual modalities. These studies have revealed a role of orbitofrontal cortex (OFC); for example, it has been shown that activity in the rat OFC reflects the degree of uncertainty in decisions based on olfactory information during reward anticipation[2]. Similar to our results for the ACC, inhibition of OFC impairs behavioral adjustments to decision confidence, but not decision accuracy[11]. However, there are several important differences, not just similarities, between previous and present studies. For example, Lak et al.[11] showed an opposing pattern for waiting time on correct and incorrect trials described above[11], whereas we found that waiting time increased for both correct and incorrect trials, across all contrast levels, and this relationship depended on the ACC. Our pattern of results also differed from that of Lak et al.[11] wherein they showed reduced waiting time on the most certain trials (greatest SNR value)[11]. Yet both our present findings and those of Lak et al. could be interpreted as a reduced variance in waiting time that indicates rats may be using an average (15 s wait time average, in the case of our experiment) in lieu of sensory evidence[11]. Importantly, unlike waiting times, reaction times did not exhibit this pattern, suggesting that it is not simply a motor timing effect, but instead a true effect on confidence readout.

It is also unlikely that ACC inactivation makes rats generally more patient, or willing-to-wait overall. Indeed, we did not observe a uniform increase in waiting times across all SNR values. It has been shown that ACC inhibition actually increases impulsivity[40], which makes this explanation unlikely for the present pattern of results. Nevertheless, here, we show that the ACC plays a similar role to the OFC but in visual information processing. That is, the ACC may guide commitment to and persistence with the current behavior based on the quality of visual evidence that led to the decision. These similarities offer interesting possibilities for the frontocortical mechanisms of confidence estimation, and suggest there may not be a subregional specialization for this process[41,42]. Consequently, future research using temporally-precise, causal manipulations are

needed to determine differential roles of ACC and OFC in decision confidence and learning under uncertainty.

We show here that unlike BLA inhibition, ACC inhibition renders confidence readout rather insensitive to both attributes of visual stimuli (SNR and contrast), suggesting that ACC gain modulates visual uncertainty computed in visual areas to determine perceptual uncertainty and post-decision processes. Anatomically, the ACC is densely interconnected with visual cortices in rodents[43,44], particularly the more rostral aspect of ACC in rat as we have targeted here[44]. Furthermore, this brain region is well positioned to integrate information about stimuli, actions, and rewards by tracking trial-by-trial outcomes of responses[45–47]. In our task, inhibition of the ACC rendered post-decision waiting times less sensitive to the strength of visual information and performance accuracy across trials, without affecting perceptual discrimination itself: i.e., impaired second order but left the first order processes intact[21]. Previous work in primates has demonstrated that confidence reports are informed by both decision difficulty and elapsed decision time (or reaction time[23,28,48]). Even in the absence of a change in decision accuracy, longer reaction times are associated with lower confidence. Here, we demonstrate that the same effect is present in rats and is also supported by the ACC. Finally, we found that ACC inhibition decreased metacognitive efficiency, or the trial-by-trial correspondence between decision accuracy and waiting times. In humans, a similar effect has been reported for perturbations of activity in the dorsolateral prefrontal cortex, which is shown to be important for visual metacognition[8].

We note that our finding on gain modulation of an uncertainty signal by ACC is not incompatible with an involvement of cingulate in sensory modulation and in contributions of sensory signals to confidence computations[48,49]. ACC inhibition could reduce center-surround modulation of visual responses which, in turn, negatively impact confidence computations. However, we should be cautious in comparing results across different techniques: microstimulation in monkeys, optogenetic activation in mice, and DREADDs inhibition in rats. Both microstimulation and optogenetic activation are more acute manipulations than a chemogenetic approach used here.

We note that waiting time is an indirect measure of confidence and as such, the effect of brain manipulations should be interpreted with caution. Firstly, several cortical and subcortical brain regions participate in reward timing[50–52]. Secondly, an overall reduction in waiting time can result from an increased delay sensitivity or impulsivity and therefore may not be reflective of confidence per se. Here, we found that inhibition of the BLA renders rats less willing to wait overall. However, this effect of BLA inhibition was independent of the strength of visual evidence to make a perceptual decision. Relatedly, there is evidence that lesions and/or pharmacological inactivation of BLA results in a delay averse (impulsive) phenotype[14] and we view the global leftward shift in rats' waiting times as consistent with those findings. Furthermore, whereas inhibition of the ACC decreased metacognitive efficiency, inhibition of the BLA failed to change this measure. Thus, during perceptual decision-making, the BLA may overall increase waiting time for reward, perhaps enabling other brain regions to interpret and/or act on ACC signals related to the strength of visual information.

Recent work documents important similarities in visual information processing between rodents and primates, although species differences do exist[53,54]. Pigmented rat strains, like the Long–Evans strain we studied here, have previously been used for vision research[54]. Here, we found that rats also show high levels of visual metacognition, adjusting post-decision waiting times based on the uncertainty in perceptual decisions. This may allow direct comparison with the modality most often assessed in

human and nonhuman primates while enabling easier, precise circuit manipulations.

We show that stimulus–response remapping is facilitated by perceptual certainty. Critically, both the BLA and ACC are required for faster learning when perceptual certainty is strong enough to improve learning. Considering that the BLA only shifts confidence readout, the observed reversals of all benefits of confidence on learning strategies and consistency in following a stimulus–response rule after BLA inhibition suggest a direct role of BLA in learning under uncertainty. That is, if the influence of BLA on learning was due to shifting confidence readout we would expect a bias in a certain direction and not reversal of all effects. In contrast, the effects of ACC seem to work through distorting confidence readout because its inhibition mainly attenuated the effect of confidence on learning.

Our results are also consistent with previous observations that the ACC–BLA circuit adjusts the levels of attention directed at environmental cues for learning based on prediction errors[45]. For example, it has been shown in rats that there is strong attention-related activity in the ACC during the entire trial following unexpected changes in reward which is most pronounced prior to and during outcome-predictive cues[45]. In contrast, unsigned reward prediction errors in the BLA may serve as attention signals, occurring at the time of unexpected reward delivery and omission[55]. Indeed, the present data prompt a revisit of prior findings on amygdala: post-training amygdala lesions have been found to facilitate reversal learning[56,57], result in enhanced learning from negative feedback during reversals[58], and lead to faster instrumental extinction[59] under fully predictive conditions. Due to extensive pretraining, these effects could be driven by a high confidence state that is slow to update without amygdala.

The ACC and BLA share direct and indirect bidirectional projections and the activity in this circuit appears to be required for adaptive learning under conditions of uncertainty in the visual cues guiding decisions or perhaps under more general cases of learning under uncertainty[60–62].

## Methods

**Subjects**. In total 31 male outbred Long–Evans rats (Charles River Laboratories, Crl:LE, Strain code: 006) were used in the experiments. The housing room in the vivarium was maintained under a reversed 12/12 h light/dark cycle at 22 °C and all behavioral testing was conducted during rats' active phase, of the dark portion of the cycle (between 08:00 and 18:00 h). Rats remained undisturbed for 3 days after arrival to our facility to acclimate to the vivarium. Each rat was then handled for a minimum of 10 min once per day for 5 days. Following handling, rats underwent stereotaxic surgery to express inhibitory DREADDs (or control null virus to express only a fluorescent protein but no mutant receptors) and were allowed to recover for three weeks. Rats were subsequently food-restricted to ensure motivation to work for food for one week prior to and during the behavioral testing, while water was available ad libitum except during behavioral testing. All rats were pair-housed at arrival and separated on the last day of handling to facilitate post-surgical recovery and minimize aggression during food restriction. We ensured that rats did not fall below 85% of their free-feeding body weight, and we saw a significant increase in rat body weight throughout the prolonged behavioral testing. On the last two days of food restriction prior to behavioral training, rats were fed 20 sugar pellets in their home cage to accustom them to the food rewards. For each experiment, rats were randomly assigned into groups, with the exception of assignment into HC and LC conditions for reversal learning as detailed below. All procedures were approved by the Chancellor's Animal Research Committee at the University of California, Los Angeles.

Viral constructs. We used inhibitory (Gi-coupled) DREADDs on a CaMKIIa promoter to transiently inactivate projection neurons in the ACC and BLA during performance on the behavioral task. An adeno-associated virus AAV8 driving the hM4Di-mCherry sequence under a CaMKIIa promoter was used to express DREADDs on putative projection neurons in the ACC or BLA (AAV8-CaMKIIa-hM4D(Gi)-mCherry, packaged by Addgene). Furthermore, a virus lacking the hM4Di DREADD gene and only containing the fluorescent tag EGFP (AAV8-CaMKIIa-EGFP, packaged by Addgene) was infused into the ACC or BLA in separate groups of animals as a null virus control.

There were four experimental groups of animals: the active DREADD virus in BLA ($n = 8$), the active DREADD virus in ACC ($n = 7$), the null EGFP virus in BLA ($n = 8$), and the null EGFP virus in ACC ($n = 8$). The DREADD and null

EGFP groups underwent identical surgeries and allowed us to control for nonspecific effects of surgical procedures and exposure to AAV8.

DREADDs are inert at baseline and can be activated by administration of Clozapine-N-oxide (CNO) to inhibit DREADD-expressing neurons. To control for nonspecific effects of injections and handling stress, we also injected animals with vehicle solution. To increase power and decrease the number of animals used in experiments, we used a within-subject design for assessing the effects of CNO, with all rats receiving CNO and vehicle injections in counterbalanced order. Inhibition of the ACC and BLA is achieved by administration of CNO to animals with DREADD expression in these brain regions. The other conditions (DREADD + vehicle, EGFP + CNO and EGFP + vehicle) served as controls for nonspecific effects of viral infusions, CNO injections, and handling stress. We confirmed there were no effects of virus exposure or nonspecific effects of CNO (see Supplementary Notes 5). This allowed us to compare the DREADDs (DREADD + CNO) effects with the DREADD + vehicle, which for brevity we refer to as vehicle or the control condition in most of the paper.

**Surgery**. All surgeries were performed using aseptic stereotaxic techniques under isoflurane gas anesthesia (5% in $O_2$ during induction and 2–2.5% in $O_2$ for maintenance). After being placed into a stereotaxic apparatus (David Kopf; model 306041), the scalp was incised and retracted. The skull was then leveled to ensure that bregma and lambda were in the same horizontal plane. Small burr holes were drilled in the skull to allow cannulae with an injection needle to be lowered into the BLA (the injection needle extended 1 *mm* below the cannulae and its tip was at AP: −2.5; ML: ±5.0; DV: −7.8 (0.1 μl) and −8.1 (0.2 μl) from skull surface) or ACC (0.3 μl, AP = +3.7; ML = ±0.8; DV = −2.6). The injection needle was attached to polyethylene tubing connected to a Hamilton syringe controlled by a syringe pump. The viruses were infused bilaterally at a rate of 0.1 μl/min. For the BLA, the ventral infusion was administered first (at −8.1) followed by the dorsal site (−7.8) since our prior experiments demonstrated more precise targeting with this approach. There was no waiting time between the two infusions for BLA. After the last viral infusions in BLA or single infusion in ACC, the needle was left in place for 10 min to allow for diffusion of the virus, after which the cannulae were slowly lifted out of the brain and the wounds stapled. Each surgery took approximately 40 min. All rats were given a 3-week recovery period prior to food restriction and subsequent behavioral training. Carprofen (5 mg/kg, s.c.) was administered for 5 days postoperatively to minimize pain and discomfort. Behavioral measures of discomfort and conditions of the wounds were monitored daily, and all surgical staples were removed within 7–10 days after surgeries depending on a rat's recovery.

**Electrophysiological confirmation of DREADDs**. Separate rats were prepared with ACC DREADDs using identical surgical procedures to the main experiments. Three rats were deeply anesthetized with isoflurane and decapitated. Slice recordings did not begin until at least three weeks following surgery to allow sufficient hM4Di receptor expression. Slice recording methods were similar to those previously published[63]. Briefly, the brain was rapidly removed and submerged in ice-cold, oxygenated (95% $O_2$/5% $CO_2$) artificial cerebrospinal fluid (ACSF) containing (in mM) as follows: 124 NaCl, 4 KCl, 25 NaHCO$_3$, 1 NaH$_2$PO$_4$, 2 CaCl$_2$, 1.2 MgSO$_4$, and 10 glucose (Sigma-Aldrich). 400-μm-thick slices containing the ACC were cut using a Campden 7000SMZ-2 vibratome. Slices from the site of viral infusion were used for inhibitory G$_i$ validation. Expression of mCherry was confirmed post hoc. Slices were maintained (at 30 °C) in interface-type chambers that were continuously perfused (2–3 ml/min) with ACSF and allowed to recover for at least 2 h before recordings. Following recovery, slices were perfused in a submerged slice recording chamber (2–3 ml/min) with ACSF containing 100 μM picrotoxin to block GABA$_A$ receptor-mediated inhibitory synaptic currents. A glass microelectrode filled with ACSF (resistance = 5–10 MΩ) was placed in layer 2/3 ACC to record field excitatory postsynaptic synaptic potentials and population spikes elicited by layer 1 stimulation delivered using a bipolar, nichrome-wire stimulating electrode placed near the medial wall in ACC. Stimulation intensity (0.2 ms duration pulses delivered at 0.33 Hz) was set to the minimum level required to induce reliable population spiking in ACC. Once reliable responses (measured as the area of postsynaptic responses over a 4-s interval) were detected, baseline measures were taken for at least 10 min, followed by a 20 min bath application of 10 μM CNO. Unless noted otherwise, all chemicals were obtained from Sigma-Aldrich.

**Behavioral training**. Behavioral training was conducted in operant conditioning chambers (Model 80604, Lafayette Instrument Co., Lafayette, IN) that were housed within the sound- and light-attenuating cubicles. Each chamber was equipped with a house light, tone generator, video camera, and LCD touchscreen opposing the pellet dispenser. The pellet dispenser delivered 45-mg dustless precision sucrose pellets. Software (ABET II TOUCH; Lafayette Instrument Co., Model 89505) controlled the hardware. During habituation, rats were required to eat five pellets out of the pellet tray inside of the chambers within 15 min before exposure to any stimuli on the touchscreen. They were then progressively trained to respond to visual stimuli presented on the screen, to initiate the trial, report the orientation of

the visual stimulus (vertical or horizontal) by nosepoking left or right on a white square stimulus, and wait for rewards.

**Behavioral testing and experimental paradigm**. A rat first initiated each trial by nosepoking a bright white square in the center of the screen. The initiation stimulus then disappeared, and a rat was briefly (1 s) presented with a vertical (V) or horizontal (H) Gabor patch embedded in noise, and required to report the orientation (H or V) based on a complementary stimulus–response rule, e.g., H → left and V → right. These spatial responses were made by nosepoking the right or left compartments of the touchscreen that became illuminated after the disappearance of the oriented visual stimulus. We altered two properties of the visual stimuli to manipulate their ambiguity. First, we changed the SNR, defined as the ratio of the contrast of the Gabor patch relative to the contrast of the added Gaussian noise. Second, we changed the overall contrast of both the Gabor patch and the added noise for a given SNR. Gratings were 200 pixels square, with spatial frequency 20 px/cycle. For training, gratings were presented at 100% contrast. For testing, gratings were embedded in white noise as follows. To create different contrasts designed to produce a range of performance (measured by $d'$) and confidence (measured by waiting time) responses such that HC and LC conditions could be established, animals performed the task on 40, 60, and 80% maximum contrast Gabor patches embedded in noise also with three possible levels of increasing contrast, for nine possible full-factorial combinations in total. This method of constant stimuli[64] facilitated selection of a pair of stimuli from these nine levels such that the animal had produced matched perceptual performance capacity ($d'$) but different waiting time in HC and LC conditions.

Correct choices were reinforced probabilistically after a randomly assigned delay: 70% of correct responses resulted in reward delivery. Time to reward delivery was drawn from an exponential distribution with mean of 8 s and on trials with no reward, the trial ended after 40 s if no reinitiation occured. Specifically, following stimulus discrimination, rats expressed their confidence by time wagering: they could wait a self-timed delay in anticipation of reward or initiate a new trial similar to previous work by the Kepecs lab[11]. The initiation stimulus appeared on the touchscreen 2 s after a rat indicated its choice. This delay was imposed to prevent nondiscriminant responding. We define the time that the animal waited before reinitiating a trial as the waiting time (see Supplementary Fig. 7 for an example and the average the distributions of waiting time).

Following fully learning the task and testing on the perceptual decision-making with re-initiation (confidence report), rats were randomly assigned to a HC- or LC condition and experienced a reversal in the stimulus–response rule. In order to determine the visual stimuli for HC and LC conditions for each rat, we selected two SNR and contrast levels that had equal discrimination accuracy ($d'$) and reinforcement history and were different only in confidence levels measured by waiting time. After determining discrimination-matched stimuli for each rat, rats were randomly assigned to LC and HC conditions and the corresponding stimulus was used for each rat based on the assigned condition. After the reversal in stimulus–response rule, rats were no longer offered an option to reinitiate the trial, but were required to wait a random delay before reward delivery or the end of the trial (on no-reward trials) following a response. This was to simplify the re-learning and ensure rats were not adopting a complex strategy due to the availability of the reinitiation option.

To study the contributions of BLA and ACC to decision-making and learning under perceptual uncertainty, we used a within-subject design: rats were given vehicle injections, CNO injections, and no injection (prior to reversal). The order of CNO and vehicle injections was counterbalanced. Therefore, a subset of behavioral sessions were preceded by inactivation of ACC or BLA pyramidal neurons via peripheral (3 mg/kg; i.p.) administration of CNO 10 min prior to the testing. The injections were administered in rats' housing room. Due to the long duration of pretraining on our task, all CNO injections were administered at least 12 weeks following the surgery, ensuring sufficient virus transduction and receptor expression. On another subset of sessions, rats received vehicle to control for behavioral effects of the stress of injections. All rats received 2-day wash-out period between drug conditions and the order of injections was counterbalanced across rats.

**Histology**. Rats were euthanized within 90 min following the last testing session with an overdose of Euthasol (Euthasol, 0.8 mL, 390 mg/mL pentobarbital, 50 mg/mL phenytoin; Virbac, Fort Worth, TX), were transcardially perfused, and their brains removed for histological processing. Brains were fixed in 10% buffered formalin acetate for 24 h followed by 30% sucrose for 5 days. To visualize hM4Di-mCherry and EGFP expression in BLA or ACC cell bodies, free-floating 40-μm coronal sections were mounted onto slides and coverslipped with mounting medium for DAPI. Slices were visualized using a BZ-X710 microscope (Keyence, Itasca, IL), and analyzed with BZ-X Viewer and analysis software.

**Signal detection theory analyses**. According to standard signal detection theory, $d'$ measures how well a subject's perceptual decisions track physical stimuli. $d'$ is preferred to other discrimination accuracy measures (i.e., percent or probability correct) because it accounts for biases such as side bias, stimulus preference[19]. Extending the same approach to confidence measures, metacognitive sensitivity

measures how well confidence tracks the likelihood that a perceptual decision is correct, and like $d'$, can also be formulated to account for bias. Specifically, Maniscalco and Lau have proposed meta—$d'$ to measure metacognitive sensitivity on the same scale as $d'$ so that one can calculate the ratio between the two (meta—$d'$/$d'$) to assess the metacognitive efficiency of a subject[31]. They defined the task of classifying stimuli as a Type 1 task, whereas the rating of confidence in this classification as a Type 2 task. Meta—$d'$/$d'$ varies between 0 and 1, where 0 indicates that the rat's trial-by-trial waiting times (i.e., confidence) do not correspond with trial accuracy and 1 indicates that the rat's Type 2 capacity is exactly matching its Type 1 sensitivity. In other words, subjects could wait a longer time before reinitiating a trial when their response is correct, and wait less time before re-initiating when the response is incorrect, considering the limitation in discrimination. Compared to commonly used Type 2 receiver operating characteristic (ROC) analysis, the meta—$d'$/$d'$ approach has the advantage of allowing one to isolate the effects of confidence on behavior from basic perceptual performance capacity. To calculate meta—$d'$, we used MATLAB (MathWorks, Natick, MA) functions freely available at http://www.columbia.edu/~bsm2105/type2sdt/.

**Data analyses**. We used MATLAB (MathWorks, Natick, MA; Version R2018b) for data and statistical analyses. For trial-by-trial learning analyses, we included data from all 15 rats that completed a total of 127,303 trials in 603 sessions. Learning occurred in a mixed design with three within-subject/repeated-measures of stage (vehicle, inhibition, no-injection) conditions and three between-subjects- (group) conditions of vehicle, ACC inhibition, and BLA inhibition. All 15 rats experienced the vehicle and no-injection conditions. However, 8 and 7 rats experienced BLA and ACC inhibition conditions, respectively. Since it was possible that a rat received reward prior to intended re-initiation of a trial, we excluded rewarded trials in the analysis of waiting time and reaction time. Furthermore, we excluded the trials in which reaction time deviated from the mean of the reaction time of the session by more than three times the standard deviation. This exclusion criterion was observed because there were trials in which the rat starts moving around and ignores the stimulus. This criterion resulted in removal of 1.1% of the trials.

Unlike the analyses presented in the preceding sections with $n = 7$ or 8 for each targeted brain region, statistical results for after reversal were restricted to 3 or 4 rats per group due to the additional HC and LC conditions ($n = 4$ rats in each of the HC and LC conditions with the BLA as the targeted region, $n = 4$ rats in the HC condition and $n = 3$ rats in the LC condition with the ACC as the targeted region). This constraint was a consequence of the experimental (within-subject) design and longitudinal nature of pretraining on the task, followed by extensive learning. Not including pretraining, the mean number of sessions to complete both the initial and relearning part of the experiments was 40–210 trials on average, in each session.

For comparisons of learning strategies and consistency following in following a stimulus–response rule across different experimental conditions, we used permutation tests. More specifically, we first calculated the actual probability of using a strategy (e.g., Win–Stay) in the observed data and then permuted this data 10,000 times to construct the permutation or null distribution. We then calculated the probability of obtaining the observed value for use of the strategy based on the null distribution from which we estimated $p$ values[65]. We used Bonferroni correction on $p$ values.

To compare the rate of learning after reversal we used an exponential function to fit the learning curve from all rats in a given experimental condition

$$P(C) = \frac{1}{1 - \exp(-\text{trial}/\alpha)}, \qquad (1)$$

where $\alpha$ is the learning parameter.

Finally, we showed the only significant difference between the HC and LC conditions prior to reversal learning was waiting time for the assigned contrast-SNR pairs that were administered after reversals. Since contrast is correlated with SNR, we used a stepwise regression to find the variables which contributed significantly in explaining the response variables (waiting time, reward history). Confidence condition was entered as a predictor variable in a GLM along with other variables (SNR, contrast, targeted brain region, and trial accuracy) in a stepwise manner to observe which of these increased adjusted R-squared significantly for the response variables.

**Rule-based repetition index (RRI)**. In order to examine the consistency in following a stimulus–response rule on two consecutive trials, we used a repetition index that was previously introduced to capture tendency to repeat the same choice beyond what is expected by chance[34], and extended it to selection based on response rules. Specifically, we computed the probability that the same rule (either correct or incorrect) was used on two consecutive trials, $p(\text{StayRule})$, and subtracted the tendency to repeat the same rule on two consecutive trials due solely to chance to arrive at the rule-based repetition index

$$\text{RRI} = p(\text{StayRule}) - p(C) * p(C) - (1 - p(C)) * (1 - p(C)), \qquad (2)$$

where $p(C)$ is the probability of choosing the correct rule. Therefore, unlike other perseveration indices[66], the RRI accounts for the probability of using the same rule on consecutive trials by chance.

**Reporting summary**. Further information on research design is available in the Nature Research Reporting Summary linked to this article.

## Data availability

All data generated and/or analyzed during this study are available in the following repository: https://gin.g-node.org/aizquie/Izquierdo_Lab_UCLA.

## Code availability

All customized behavioral task codes can be found on https://github.com/izquierdolab/perceptual-uncertainty.

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

## Acknowledgements

We thank Dr. Brian Maniscalco for helpful comments on these data. This work was supported by CRCNS R01DA047870 (Soltani and Izquierdo), UCLA's Division of Life Sciences Recruitment and Retention Fund, and UCLA Academic Senate Grant (Izquierdo). We acknowledge the Charles E. and Sue K. Young Fellowship (Stolyarova). We also thank the Staglin Center for Brain and Behavioral Health for enabling the use of equipment for microscopy imaging.

## Author contributions

A. Stolyarova, MAKP, H.L. and A.I. conceived and designed the study; A. Stolyarova, E.E.H. and T.J.O. collected the data and analyzed first set of results; M.R., A. Soltani. provided new data analyses; M.R. implemented data analyses and prepared the figures; A. Stolyarova, M.R., E.E.H., T.J.O., MAKP, A. Soltani and AI interpreted the results of analyses; A. Stolyarova, M.R., A. Soltani, and A.I. wrote the first draft; M.R., A. Soltani and A.I. revised the paper; all authors contributed to editing of the paper.

## Competing interests

The authors declare no competing interests.
