## [Peer Review File · Nature Communications]

Reviewers' Comments:

Reviewer #1:

Remarks to the Author:

The manuscript examines the effect of ACC or BLA inactivation on perceptual decision confidence. To do so, the manuscript uses a behavioral paradigm established in previously published paper, in which rats were trained to wait for the reward after making a perceptual choice. The manuscript extends this to visual decisions in rats, and uses DREADDs to inactivate ACC or BLA during the task. The overall conclusion is that, during the established behavior, ACC inactivation impairs confidence judgment but BLA inactivation decreases overall wait time. During the reversal learning both structures seem to contribute to task performance. Please find my issues and suggestions below:

1. It would be crucial to see rat's psychometric curves in the task before looking into their waiting behavior, in particular when it comes to thinking about error trials. Second bets (or waiting time in this case) could work as the proxy of confidence only if psychometric curves are robust and approach almost perfect performance for easy stimuli (and hence the source of noise in decisions could be attributed, largely, to imperfect perception). Without psychometric plots it is hard to distinguish whether errors reflect wrong perception or factors such as attention etc, making it near-impossible to interpret waiting time behavior.
2. The fact that in the current experiments waiting time significantly increases as a function of stimulus (or SNR) in error trials indicates that waiting time in these trials is not a proxy for confidence. The inverse relation between confidence and sensory evidence in error trials comes out of various normative frameworks such as Bayes rule (as various papers have shown), and it is a defining signature of confidence. As such, the fact that waiting times increases in error trials suggests that interpreting waiting time in these trials is not straightforward; they might reflect confidence plus various other factors (see below). This is making it absolutely essential to visualize psychometric curves before looking into waiting behavior.
3. Related to the point about waiting time in error trials, in the discussion the manuscript suggests that the reason for the increase of waiting times in the error trials (opposite to previous papers) is due to the tested sensory modality (visual stimuli having short transient while olfactory stimuli having longer temporal profile). However, there are several studies using auditory or visual stimuli in human and primates which showed that in error trials confidence rating decreases as a function of sensory evidence. Thus, I am finding it hard to think the sensory modality is the reason, and I think that in the current experiments error trials are not only due to wrong perception and hence waiting times do not only reflect confidence.
4. One could think of various reasons for apparent increased waiting for error trials with higher SNR. One of them could be that errors for high SNR happen more frequently in the later trials of the session. This is when the motivational state of the animal is relatively lower, and it thus, on average, reports longer waiting time. This brings up the questions of whether errors were distributed evenly throughout the session? and whether waiting times grew as a function of trial number? Moreover, given the variability of average waiting time across sessions (see below), it could be that errors for high SNR stimuli are more frequent in sessions in which average wait time is high, and hence high wait time for errors with high SNR.
5. Many of the analysis on waiting time and RTs report the average waiting time per session (for example Fig. 2a,b and others), and they do so on the raw data. These left me with an issue of huge variability of RTs and waiting times across days (for example for say SNR = 4, the average wait time could be between 8 to 20 s in different days). The first question is there is such a huge variability?

Were animals still learning? Having this level of variability, I think the more sensible way to look into the wait time and RT is to perform session-by-session normalization (such as z-scoring) before averaging. This would be important for the RT data because the manuscript argues that RTs are not reflective of confidence. I suspect that the difference between RTs of error and correct trials will appear once the analysis is accounting for the session-by-session variability. This analysis will also remove some of the issues about the wait time in error trials (point 4).

6. Apart from example histology in figure 1, I did not see any quantification of the anatomical region of inactivation across animals. It would be useful to see further figures for that, and importantly, it would be useful to see if there is any relation between the exact position of inactivation and the type of behavioral effect. This is in particular important because recent works using somatosensory, auditory or visual stimuli in rodents are reporting impaired perceptual performance in frontal regions very close to ACC (perhaps slightly more dorsal).

7. The manuscript concludes that ACC inactivation impairs confidence judgment but BLA inactivation decreases the overall wait time. The difference between these regions could be due to different efficiency in inactivating these regions. Was there any procedure in titrating the level of inactivation? Could it be that with different level of inactivation BLA and ACC results appear similar (like stronger BLA inactivation could result in an effect on confidence estimates?) Electrophysiological confirmation of efficiency of inactivation is only tested in ACC and not in BLA.

8. I found it very hard to interpret the reversal learning experiment. This is partly because of the issues above, and also the fact that animals were suddenly not allowed to reinitiate trials in this experiment. Were the animals learning two things (not to reinitiate and reverse) at the same time? In the same section in Line 403, it is stated that both LC and HC groups received the same amount of reward in the session. I thought HC group learned faster (and hence more rewards)? I then thought that LC groups have shorter wait time and hence larger number of trials and possibly more rewards over time? These needs further analysis and clarifications.

9. Overall, I found it very hard to read the paper. Part of this was due to the fact that the text is buried in between long parentheses dedicated to stat test reports. I counted more than 15 instances in which the stat report is extended over 2-3 lines and often followed immediately by another long stat report. This made it very hard for me to follow the paper in the current format.

Minor:

- Line 183 refers to Figure 2A inset. I do not see an inset there.
- The analysis in Figure 4C requires further clarification.
- The section looking into the effect of confidence on reversal learning; it is dividing animals to low and high confidence groups. After looking into Methods, I still left unsure how exactly these two groups are defined and matched.

Reviewer #2:

Remarks to the Author:

The manuscript describes the first study in rodents to show how confidence, as measured by post-decision waiting times, affects both perceptual inference and learning. Moreover, they show, using DREADDs, that inhibition of anterior cingulate cortex (ACC) and not the amygdala (BLA) disrupts signal-to-noise ratio (SNR) effects on confidence, whereas inhibition of the BLA completely shifts

perceptual certainty to slow down reversal learning. This is excellent work and I only have few minor comments:

The finding that high confidence in the stimulus-response rules facilitates reversal learning is intriguing given evidence that lesions of the amygdala and interconnected regions, including ACC, appear to lower evidence thresholds for learning stimulus-outcome reversals and concomitantly strengthen prior beliefs that learned relationships are unstable (Jang et al., 2015). This result could not be attributed to differences in win-stay or lose-shift performance. One explanation, given the results of this manuscript, is simply that the monkeys with amygdala and ACC lesions are less confident about their learning. I would be interested in hearing the authors speculate about their results relate to this prior study, and more generally studies that have observed enhanced reversal learning in monkeys and rodents following amygdala lesions (e.g. Stalnaker et al. 2007; Rudebeck et al., 2008). For example, could cueing reversals based on reaching a predetermined criterion represent a high confidence state and are most reversal learning studies biased in this sense? Might a high confidence state also be a way of interpreting the senior author's own finding that monkeys extinguish faster (e.g. Izquierdo and Murray, 2005)?

The distribution of the waiting times doesn't necessarily match the distribution of the reward delivery times, at least over the very short intervals. The waiting time distribution appears Poisson distributed, whereas the reward times follow an exponential distribution. This would seem to imply that the animals are waiting up to a certain minimum threshold. What do the authors think might account for that threshold and is it possible related to some aspect of the task?

Given that the ACC and BLA manipulations were in different groups of animals it would be helpful to plot the vehicle effects for each group (Fig 3 and S3). I am wondering to what extent in the contrast = 80 condition if averaging the two vehicle groups is possibly masking what is a general effect for the ACC inhibition that resembles that seen in the contrast=40 and contrast=60 conditions. In other words is are the vehicle waiting times for the BLA group especially long in the contrast=80 group.

With respect to the odd effects of contrast on the waiting times, is this possibly due to individual differences in perception among the rats tested? Instead of averaging across the three contrast levels, and effectively ignoring them, would it be possible to try and determine high and low contrast possible using d' as a sorting metric? I'm curious because the results look very similar between the 40 and 60 conditions, and I am wondering if the non-linear effects in the 80 condition represent hitting some perceptual threshold in some animals and not others.

I would be interested to see the correlation for the reaction and waiting times broken out by correct and incorrect trials. Also, are the RTs plotted in Figure 2 pooled across all of the animals? Why not simply plot a histogram of the within session correlations as that is reported in the results.

The double-y plots in Figures 2 are quite confusing and not well explained in either the figure legend or text. What elements relate to each of the y-axes? Perhaps it would be simpler to just break out the plots individually as is done in Figure 3?

In line 182 I believe you are referring to the inset in Figure 2D and not 2A, where there is no insert?

Reviewer #3:

Remarks to the Author:

Stolyarova, Rakhshan et al. used chemogenetic inactivation of the ACC and BLA during a temporal-

wagering perceptual decision-making task designed to measure decision confidence in rats. In this task, increased waiting times for variable-delay rewards are considered to be a measure of decision confidence. The authors report that BLA inactivation decreased waiting times by similar amounts regardless of stimulus strength and SNR, while ACC inactivation increased waiting times in a fashion that rendered them insensitive to the strength of evidence. Moreover, they show that in rats higher confidence can lead to faster learning of reversed stimulus-reward associations, and that either BLA or ACC inactivation abolishes the effect of higher confidence, albeit by changing behavioral strategies in different ways. The authors conclude that only ACC activity is required to compute confidence on a trial-by-trial basis, but that both ACC and BLA are required for learning that relies on decision confidence.

The questions posed by this study are interesting and the results are novel and of interest to the community. Moreover, the manuscript is clearly written. However, my enthusiasm is decreased by technical and interpretational issues that I believe should be addressed before I can recommend the manuscript for publication.

Major points

1. Perhaps I missed it, but I could not find a crucial inactivation control, namely CNO delivery in animals not expressing DREADDs (e.g. injected with the EGFP construct). While the other controls included in the study are important, they do not address the possibility that the effects of inactivation are not (or at least not exclusively) mediated by silencing of the neurons expressing DREADDs, but by effects of clozapine on its target receptors. CNO has been shown to cross the blood-brain barrier with very low efficiency, such that the effects of delivering it systemically are mostly due to its byproduct clozapine entering the brain and both activating DREADDs and other target receptors (Gomez et al, Science, 2017; Mahler and Aston-Jones, Neuropsychopharmacology, 2018). The authors may argue that the effects are different in BLA and ACC animals, but that could still be due to the interaction of DREADDs activation and that of other receptors that clozapine binds. The slice recordings also do not address this possibility since the blood-brain barrier is not a variable there.

2. Some puzzling aspects of task performance are not explained in the manuscript, specifically:
2.1. It seems from figure S1 that performance does not increase monotonically with contrast. For example, for SNR = 4, performance is higher for 60% than 80% contrast. It would be helpful to show a panel in this figure of performance (d') as a function of contrast and SNR, as it is hard to extract that information from panel B. Related to this, was the luminance of the screens gamma-corrected?
2.2. In Lak et al, 2014, waiting times in error trials decrease with stimulus evidence strength, which is predicted by their normative model. However, in the present manuscript, the opposite trend is seen in the data. The authors discuss that this could be due to differences in sensory modality and food vs water reward, but it is also possible that the present task structure imposes different normative constraints. Have the authors tried fitting Lak et al's model to their behavioral data, or can they provide alternative models?

3. The authors report that ACC, but not BLA, inactivation abolishes sensory-evidence dependence of wait times and conclude that this is a signature of ACC's role in confidence computations. However, the waiting times as increased across all SNRs with ACC inactivation, mimicking increased decision confidence. It seems to me that the most likely effect of removing confidence computations would be to decrease waiting times for high-confidence trials (like the effects of OFC inactivation in Lak et al). This is true both from a normative standpoint (assuming rats try to maximize reward rate), and from the empirical observation that rats tend to be impulsive. Applying similar logic to the authors' explanation of the BLA results, an alternative explanation for these findings is that inhibiting ACC has

a non-specific effect, e.g. decreases impulsivity, and that the loss of dependence on SNR is an artifact coming from a ceiling effect, i.e. that waiting times cannot be longer because of task design.

The authors discuss that these ACC findings are compatible with a role for this structure in gain-modulating uncertainty signals, but this claim is incompatible with the effects of microstimulating sensory areas on decision confidence (Fetsch et al, *Neuron*, 2014), and with the known effects of stimulating ACC on visual responses (Zhang et al, *Science*, 2014).

Along the lines of my point 2.2, fitting a normative model may help disambiguate these possibilities, as would repeating the experiments with longer reward delays.

Minor points

1. Figure 1C. Maybe it's an artifact of my pdf, but the histology figures are hard to see. Can the authors provide higher-resolution figures? Also, please add scale bars. Finally, can the authors add quantification of the location of the injection sites, as well as injection spreads? This is important for interpretation of the results, since the injections could encompass neighboring structures.
2. A few statements are not backed by references, for example lines 175 and 210. Please add.
3. The finding that ACC inhibition does not affect discrimination performance (Fig 4) seems surprising, given many previous findings in multiple sensory modalities that mPFC/ACC/M2 activity manipulations do affect perceptual decision-making performance in rodents (e.g. Rodgers and DeWeese, *Neuron*, 2014; Zhang et al., *Science*, 2014; Pinto and Dan, *Neuron*, 2015; Hanks et al., *Nature*, 2015; Goard et al., *eLIFE*, 2016; Huda et al., *bioRxiv*, 2018). Can the authors add a discussion about potential reasons?
4. Fig 4C. The more rigorous statistical approach here would be to carry out an ANOVA first, followed by post-hoc tests.
5. The results in Figure 5A are very interesting and the control analyses presented in the paragraph starting in line 392 are important. To help readers, I would suggest adding panels to Figure 5 or a supplemental Figure showing the results of these analyses, rather than just reporting them in the text.
6. Figures 5B, 6B, 6D. Please add error bars / confidence intervals. Also, did the authors correct for multiple comparisons when calculating significance for these tests?
7. Figure 6A. This is related to my major point #3 — is this incompatible with the results in Figure 3? In other words, if ACC inactivation mimics the effect of increased confidence, wouldn't one expect faster learning? Maybe this is a naive interpretation but I would recommend expanding the discussion on this point (beyond line 659-660).
8. Methods, line 850. Can the authors justify why these trials were excluded?
9. Supplementary analyses (page 33). I suggest showing these as figures instead of text to help the reader.

Response to reviewers' comments, and summary of changes made in response to their comments.

Title: Dissociable roles for Anterior Cingulate Cortex and Basolateral Amygdala in Decision Confidence and Learning under Uncertainty

Authors: Stolyarova, Rakhshan, Hart, O'Dell, Peters, Lau, Soltani, and Izquierdo

We are thankful to the reviewers for their careful reading of our manuscript and for their constructive suggestions. We have performed additional analyses and made substantial changes to the revised manuscript to address all of their concerns and suggestions. Specifically, we addressed the concern about waiting time increasing with stimulus strength on error trials (raised by Reviewers # 1 and 3) by providing experimental and theoretical evidence from other groups that observed the same pattern, and explained this pattern based on normative models. In addition, we have provided more data and information on DREADDs expression and appropriate DREADDs controls (raised by all reviewers). Below, please find our point-by-point response to reviewers' comments. The corresponding changes have been clearly marked (in blue) and noted in the revised manuscript (e.g., [R1.1] indicates response to Reviewer #1's first point).

Reviewer comments

Reviewer #1

Reviewer #1 (Remarks to the Author):

"The manuscript examines the effect of ACC or BLA inactivation on perceptual decision confidence. To do so, the manuscript uses a behavioral paradigm established in previously published paper, in which rats were trained to wait for the reward after making a perceptual choice. The manuscript extends this to visual decisions in rats, and uses DREADDs to inactivate ACC or BLA during the task. The overall conclusion is that, during the established behavior, ACC inactivation impairs confidence judgment but BLA inactivation decreases overall wait time. During the reversal learning both structures seem to contribute to task performance. Please find my issues and suggestions below:"

Response: We thank the reviewer for the time and effort in the evaluation of our work, and hope that our responses here and corresponding changes in the revised manuscript address their concerns and questions.

Major issues:

1. "It would be crucial to see rat's psychometric curves in the task before looking into their waiting behavior, in particular when it comes to thinking about error trials. Second bets (or waiting time in this case) could work as the proxy of confidence only if psychometric curves are robust and approach almost perfect performance for easy stimuli (and hence the source of noise in decisions could be attributed, largely, to imperfect perception). Without psychometric plots it is hard to distinguish whether errors reflect wrong perception or factors such as attention etc, making it near-impossible to interpret waiting time behavior."

Response: We thank the reviewer for pointing out concerns about waiting time reflecting factors other than confidence. To address these concerns, we further analyzed the effect of contrast and SNR on accuracy. First, we separated the trials by contrast and computed accuracy as a function of SNR. We observed that accuracy increases as SNR increases for each of the contrast levels (**Fig. 1A** below; GLM contrast=40: $p = 1.14 \times 10^{-27}$; $\beta_{SNR} = 0.108$, $p = 1 \times 10^{-3}$; GLM contrast=60: $p = 6.07 \times 10^{-32}$; $\beta_{SNR} = 0.157$, $p = 2.08 \times 10^{-8}$; GLM contrast=80: $p = 1.19 \times 10^{-11}$; $\beta_{SNR} = 0.087$, $p = 2 \times 10^{-3}$). Observing the effect for each contrast, we also calculated accuracy as a function of SNR across all contrast values. Similarly, we found that accuracy increases with SNR on accuracy (**Fig. 1B** below; GLM: $p = 5.01 \times 10^{-70}$; $\beta_{SNR} = 0.118$, $p = 1.23 \times 10^{-11}$). These results illustrate that for easier stimuli (higher SNR), performance is better compared to the more difficult stimuli (lower SNR) as mentioned by the reviewer.

Although we would have liked to have had a wider range of possible SNR and contrast values, the training and testing duration for the present number of stimuli was already exceedingly long (approximately 6 months) for the rats. Furthermore, although a highly discriminable (easy) stimulus may result in nearly perfect performance, it cannot completely rule out that uncertainty during a near threshold condition is due to external noise. Indeed, there are studies that do not have such stimuli in the experiment while exploring confidence (e.g., Hertz, Bahrami, & Keramati, 2018).

We also do not suggest that uncertainty is driven entirely by internal noise. Instead, we suggest that there is indeed perceptual uncertainty, in part driven by the stimuli themselves. Further, we show that given the same stimulus contrast level and SNR, waiting time predicts accuracy, suggesting there is at least monitoring of the success of the perceptual process.

In the revised manuscript, we now have clarified this point by providing the statistics for accuracy as a function of SNR and contrast in the **Supplementary Analysis 1**. In addition, we have added **Fig. 1** below as new **Supplementary Figure 1** (see [R1.1] in the revised manuscript).

Figure 1. Accuracy increases for easier stimuli. (A) Accuracy is plotted as a function of SNR for different values of stimulus contrast. The squares and error bars show the mean and S.E.M. The solid lines show the fitted psychometric curve over the observed data. Different values of contrast are indicated in the legend and correspond to least discriminable (contrast = 40), moderately discriminable (contrast = 60), and most discriminable (contrast = 80). (B) Plots shows the average accuracy as a function of SNR across all contrast levels.

2. *“The fact that in the current experiments waiting time significantly increases as a function of stimulus (or SNR) in error trials indicates that waiting time in these trials is not a proxy for confidence. The inverse relation between confidence and sensory evidence in error trials comes out of various normative frameworks such as Bayes rule (as various papers have shown), and it is a defining signature of confidence. As such, the fact that waiting times increases in error trials suggests that interpreting waiting time in these trials is not straightforward; they might reflect confidence plus various other factors (see below). This is making it absolutely essential to visualize psychometric curves before looking into waiting behavior.”*

Response: We thank the reviewer for raising the concerns about one of our observations that waiting time increases with stimulus strength on error trials similarly to correct trials. As the reviewer points out, many studies have used the opposing relation between confidence and sensory evidence in correct and error trials as a signature of confidence report. However, there are recent experimental and theoretical studies showing that confidence can also increase on error trials with larger sensory evidence and thus, suggest that the opposing pattern on correct and error is not necessarily the *defining* signature of confidence as previously assumed (Kiani, Corthell, & Shadlen, Neuron 2014; Rausch, Hellmann, & Zehetleitner, Attention, Perception, and Psychophysics 2018; van den Berg et al., eLife 2016).

For example, Kiani and colleagues (2014) showed that in a random-dot motion discrimination task in which subjects report choice and certainty simultaneously, certainty grows with stimulus strength for both correct and error responses (see their Figure 4). This

violates the opposing pattern of confidence on correct and error trials, which is referred to as the “divergence” pattern of confidence (Adler & Ma, Neural computation 2018). In order to explain the violation of the divergence signature of confidence based on normative models, they used a modified version of a bounded accumulation model. In this model, confidence in choosing an option is a function of both the amount of accumulated evidence supporting that option and the accumulation time. In another study, van den Berg and colleagues (2016) used a motion discrimination task to show that when the subjects are more confident, they tend to be more accurate. Subjects were more confident on correct compared with error trials but more importantly, the subjects were more confident with stronger motion coherence even on error trials. The authors used a bounded accumulation mechanism to explain how the time to decision influences confidence and why confidence in error trials increases with stronger motion coherence. Finally, Rausch et al. (2018) used two sets of experiments in which the subjects reported the orientation of a grating while the stimulus was masked with noise. They observed that confidence in both correct and error trials increase with the stimulus strength. They further proposed a probabilistic normative model (weighted evidence and visibility model) to account for the observed pattern of confidence report.

Importantly, there are also recent studies based on Bayesian framework that argue the divergence signature of confidence disappears under different conditions of stimuli, noise, and task structure (Adler & Ma, Neural computation 2018; Rausch & Zehetleitner, bioRxiv 2018). For example, Adler and Ma (2018) showed that the divergence signature of Bayesian confidence is present if category-conditioned stimulus distributions are uniform, independent of high- and low-noise regimes. Critically, they show that if the stimulus distributions are Gaussian and measurement noise is relatively low compared to the stimulus distribution width, then the divergence signature is not expected.

Similarly, by extending Bayesian inference to include evidence about stimulus strength, Rausch & Zehetleitner (2018) illustrated that the divergence signature is a special case of confidence signatures when the evidence about stimulus strength is omitted from the calculation of objective confidence (i.e., only stimulus identity is available). More specifically, they found that previous studies that observed a violation of the divergence signature of confidence used tasks in which the stimulus was composed of a feature that assessed the correct response along with an orthogonal manipulation of stimulus strength. In fact, this is the case in our experiment in which we assessed accuracy in perceptual judgment by changing SNR and contrast of the stimuli as two orthogonal manipulations.

Together, recent studies reviewed above clearly demonstrate that orthogonal components of the stimuli and the structure of the confidence report (e.g. simultaneous choice and confidence in choice vs. a two-step task) can change the signature of confidence.

Therefore, we think observing the violation of the divergence signature of confidence in our experiment is a feasible outcome based on the aforementioned studies.

In the revised manuscript, we now have devoted a separate section to discussing the aforementioned studies to address this point (see [R1.2]).

3. *“Related to the point about waiting time in error trials, in the discussion the manuscript suggests that the reason for the increase of waiting times in the error trials (opposite to previous papers) is due to the tested sensory modality (visual stimuli having short transient while olfactory stimuli having longer temporal profile). However, there are several studies using auditory or visual stimuli in human and primates which showed that in error trials confidence rating decreases as a function of sensory evidence. Thus, I am finding it hard to think the sensory modality is the reason, and I think that in the current experiments error trials are not only due to wrong perception and hence waiting times do not only reflect confidence.”*

Response: Thanks to the reviewer for raising this concern. We agree that there could be other and more important reasons for the observed increase in waiting time as a function of evidence in error trials. Therefore, we have removed this speculative statement (see [R1.3]), and now discuss our findings in the light of literature detailed in the previous comment.

4. *“One could think of various reasons for apparent increased waiting for error trials with higher SNR. One of them could be that errors for high SNR happen more frequently in the later trials of the session. This is when the motivational state of the animal is relatively lower, and it thus, on average, reports longer waiting time. This brings up the questions of whether errors were distributed evenly throughout the session? and whether waiting times grew as a function of trial number? Moreover, given the variability of average waiting time across sessions (see below), it could be that errors for high SNR stimuli are more frequent in sessions in which average wait time is high, and hence high wait time for errors with high SNR.”*

Response: Thanks to the reviewer for raising this concern. In addition to several explanations for an increased waiting time on error trials provided above, we have performed additional analyses to address the reviewer’s concerns about other possibilities that could account for our results. First, we examined the effect of trial number on accuracy for high SNR (SNR = 4) and found no significant effect (GLM: $p = 4.73 \times 10^{-11}$; $\beta_{trial} = 0.001, p = 0.19 (> p^* = 0.05/3)$). Similarly, we did not find a significant effect of trial number on accuracy considering all SNR values together (GLM: $p = 7.05 \times 10^{-157}$; $\beta_{trial} = -0.0002, p = 0.9$). These analyses thus do not provide any evidence for differential distribution of errors throughout the session. Second, we found that z-scored waiting time did not significantly increase for high SNR as a function

of trial number (GLM: $p = 3.11 \times 10^{-192}$; $\beta_{trial} = -0.0003, p = 0.8$). In addition, z-scored waiting time did not significantly increase as a function of trial number within a session (GLM: $p = 1 \times 10^{-170}$; $\beta_{trial} = -0.0016, p = 0.49$). Therefore, we did not find any evidence that motivational state (which could decrease with time and satiation) influences accuracy. Finally, we did not find any significant correlation between average waiting time and accuracy for high SNR (Pearson correlation, $r = -0.06, p = 0.8$). Together, our analyses illustrate that the observed increase in waiting time (confidence) on error trials with high SNR stimuli is likely not due to uneven distribution of errors throughout the session or due to a low motivational state.

We have included above results to the **Supplementary Analysis 6** in the revised manuscript to address the reviewer's concern (see [R1.4]).

5. *"Many of the analysis on waiting time and RTs report the average waiting time per session (for example Fig. 2a,b and others), and they do so on the raw data. These left me with an issue of huge variability of RTs and waiting times across days (for example for say SNR = 4, the average wait time could be between 8 to 20 s in different days). The first question is there is such a huge variability? Were animals still learning? Having this level of variability, I think the more sensible way to look into the wait time and RT is to perform session-by-session normalization (such as z-scoring) before averaging. This would be important for the RT data because the manuscript argues that RTs are not reflective of confidence. I suspect that the difference between RTs of error and correct trials will appear once the analysis is accounting for the session-by-session variability. This analysis will also remove some of the issues about the wait time in error trials (point 4)."*

Response: We thank the reviewer for suggesting a session-by-session normalization of waiting time and reaction time. We have now repeated our main analyses using z-scored waiting and reaction times (see below). However, we note that animals were well-trained on the task and variability did not likely reflect ongoing learning, but rather that it reflected the range of individual rat performance. In tasks involving either timing or effort, there are typically stable individual differences observed in rats (broadly we have observed this to be the case in our experiments, but also see the work of Catharine Winstanley). Indeed, as mentioned above, we did not find any evidence that errors were differently distributed throughout the experiment (GLM: $p = 7.05 \times 10^{-157}$; $\beta_{trial} = -0.0002, p = 0.9$). Additionally, z-scored waiting time did not significantly change throughout the session (GLM: $p = 1 \times 10^{-170}$; $\beta_{trial} = -0.0016, p = 0.49$).

To fully address the reviewer's concern, we repeated our main analyses using z-scored waiting time and reaction time. First, we found that z-scored waiting times were longer on trials with a larger SNR (GLM: $p = 5.68 \times 10^{-242}$; $\beta_{SNR} = 0.23, p = 1.84 \times 10^{-24}$; **Fig. 2A**

below). In contrast, z-scored reaction time decreased with increased SNR (GLM: $p = 5.89 \times 10^{-224}$; $\beta_{SNR} = -0.46$, $p = 2.83 \times 10^{-81}$; **Fig. 2B** below). Additionally, z-scored waiting times were negatively correlated with z-scored reaction times on a trial-by-trial basis (GLM: $p = 1.98 \times 10^{-161}$; $\beta_{RT} = -0.21$, $p = 1.98 \times 10^{-161}$; **Fig. 2C** below).

To examine the relationship between accuracy and time wagering, we also computed the waiting time for correct and incorrect responses. We found that rats waited significantly longer following correct relative to incorrect responses, or trial type (diff(mean)=0.58; GLM: $p = 1.83 \times 10^{-78}$; $\beta_{trial\ type} = 0.56$, $p = 6.24 \times 10^{-41}$; **Fig. 2D** below). In addition, waiting times on both correct and incorrect responses increased with greater SNR (GLM Correct: $p = 7.81 \times 10^{-102}$; $\beta_{SNR} = 0.45$, $p = 6.95 \times 10^{-58}$; GLM Incorrect: $p = 3.09 \times 10^{-48}$; $\beta_{SNR} = 0.23$, $p = 5.39 \times 10^{-28}$). These results demonstrate that not only is z-scored waiting time sensitive to the strength of the visual information, but it also reflects rats' accuracy in discrimination.

Finally, we also found that z-scored reaction times decreased with larger SNR and tended to be faster for correct responses relative to incorrect responses/discrimination (diff(mean)=-0.08; GLM: $p = 0.06$; $\beta_{SNR} = -0.094$, $p = 0.03$; **Fig. 2E** below). Overall, the results based on z-scored waiting time and reaction time yielded an identical pattern to our previous results. Importantly, this also holds true for the results of inhibition of ACC and BLA (see **Fig. 3** below).

In the revised manuscript, we have now added the point that session-by-session normalization of waiting time and reaction time result in a similar interpretation (see [R1.5]). However, for the sake of brevity and clearer presentation, we decided to keep the main data presentation (**Figures 2, 3** in the manuscript) as before and report the results based on the z-scored values in the **Supplementary Analysis 2**.

Figure 2. Waiting time serves as a proxy for confidence that is more sensitive than reaction time. (A) Waiting time before re-initiation increases as SNR increases. Plotted are the distributions of session-by-session normalized waiting times for each SNR: 2 (blue), 3 (green), and 4 (red), following vehicle administration. Solid lines show the median of each distribution. The y-axis on the right shows the probability of a correct response. Solid circles indicate the calculated probability of correct responses for each bin of the waiting time distribution, for each SNR. The dashed lines show the regression line of the probability of correct responses for bins in each SNR. (B) Reaction time decreases as SNR increases. Plotted are the distributions of reaction time for all trials z-scored for each session. Similar to panel A, y-axis on the right shows the probability of a correct response. Solid circles indicate the calculated probability of correct responses for each bin of the reaction time distribution, for each SNR. Conventions are the same as in panel A. (C) Waiting time before re-initiation of a new trial is negatively correlated with reaction time to make a choice. Z-scored waiting time is plotted as a function of the z-scored reaction time for all trials and all rats. Each data point is a trial in a session following vehicle administration. (D) Waiting time is larger for correct compared to incorrect responses for any SNR. Plotted is waiting time for all trials (black), correct trials (green), and incorrect trials (purple) for different SNR z-scored for each session. Error bars show the S.E.M. over sessions (typically smaller than the symbol). (E) Reaction time only weakly reflects the accuracy of the response. Plotted is the reaction time for all trials and separately for correct and incorrect responses z-scored for each session. Conventions are the same as in panel D. Overall, response accuracy is reflected in waiting time an order of magnitude better than in reaction time.

Figure 3. ACC and BLA inhibition result in opposing changes in waiting time, but no change in reaction times. (A) Waiting time is modulated differently for different SNRs following ACC inhibition. Plotted are the distributions of the z-scored waiting time separately for each SNR, following ACC inhibition. Solid lines show the median of each distribution. Dashed lines show the median of the same condition but following vehicle administration as is shown in Fig. 2A above. (B) Waiting time's sensitivity to SNR following BLA inhibition is similar to that of following vehicle administration. Conventions are the same as in panel A. (C) For all trials discarding the accuracy of the trial, as is shown in panel A, inhibition of ACC renders waiting times insensitive to the strength of sensory signal. Whereas as is shown in panel B, BLA inhibition preserves the sensitivity to the strength of sensory signal. Plotted is the z-scored waiting time for all trials as a function of SNR following vehicle administration (yellow), inhibition of BLA (orange), and inhibition of ACC (blue). (D-E) Same as in panel C but only on trials in which a correct (D) or incorrect (E) response was made. Error bars show the S.E.M. over sessions (typically smaller than the symbols). (F) Reaction time is unaffected by inhibition of either ACC or BLA. Plotted is the z-scored reaction time for all trials as a function of SNR following vehicle administration (yellow), inhibition of BLA (orange), and inhibition of ACC (blue). (G-H) Same as in panel F but only on trials in which a correct (G) or incorrect (H) response was made.

6. “Apart from example histology in figure 1, I did not see any quantification of the anatomical region of inactivation across animals. It would be useful to see further figures for that, and importantly, it would be useful to see if there is any relation between the exact position of inactivation and the type of behavioral effect. This is in particular important because recent works using somatosensory, auditory or visual stimuli in rodents are reporting impaired perceptual performance in frontal regions very close to ACC (perhaps slightly more dorsal).”

Response: We thank the reviewer for asking about this very important validation and quantification. To address this concern, we have produced additional photomicrographs and reconstructed the DREADDs expression. As shown in **Fig. 4** below, the expression of both DREADDs and null virus eGFP was well-targeted and we did not observe any missed targets. We note, however, that it is very rare to find reports in the literature of significant correlations between behavior and degree of DREADDs expression, spread of infusion, or even lesion extent, etc. Yet we agree with the reviewer that quantification and report of these analyses should be the gold standard method. We analyzed the effect of DREADDs spread on z-scored waiting time and accuracy following vehicle administration. We observed no significant effect of DREADDs spread on waiting time or accuracy (GLM waiting time: $p = 10^{-16}$; $\beta_{spread} = 0.4, p = 0.085$; GLM accuracy: $p = 1.65 \times 10^{-108}$; $\beta_{spread} = -0.26, p = 0.32$). Similarly, following CNO administration, we found no significant effect of DREADDs spread on waiting time or accuracy (GLM waiting time: $p = 10^{-16}$; $\beta_{spread} = -0.03, p = 0.89$; GLM accuracy: $p = 9.46 \times 10^{-120}$; $\beta_{spread} = -0.31, p = 0.24$). Together, these analyses do not provide any evidence that the spread of virus (null or active) influenced waiting time or accuracy in the task, perhaps because the minimum amount of spread inhibited the majority of the targeted brain area (ceiling effect).

We now have included these analyses in **Supplementary Analysis 9** and the **Fig. 4** below as the new **Figure 1** in the revised manuscript (see [R1.6]).

Figure 4. Expression of inhibitory DREADDs and null virus EGFP in Anterior Cingulate Cortex and Basolateral Amygdala. (A) *Top*, Representative expression of hm4Di-mCherry DREADDs under CaMKIIa is shown for ACC (left) and BLA (right). *Bottom*, Reconstructions of maximum viral spread for all rats. Numerals depict +Anterior-Posterior (AP) level relative to Bregma. Scale bars 1000 μm with 100 μm (inset). (B) *Top*, Representative expression of eGFP (null virus) under CaMKIIa is shown for ACC (left) and BLA (right). *Bottom*, Reconstructions of maximum viral spread for all rats. Numerals depict +Anterior-Posterior (AP) level relative to Bregma. Scale bars 1000 μm with 100 μm (inset).

7. "The manuscript concludes that ACC inactivation impairs confidence judgment but BLA inactivation decreases the overall wait time. The difference between these regions could be due to different efficiency in inactivating these regions. Was there any procedure in titrating the level of inactivation? Could it be that with different level of inactivation BLA and ACC results appear similar (like stronger BLA inactivation could result in an effect on confidence estimates?) Electrophysiological confirmation of efficiency of inactivation is only tested in ACC and not in BLA."

Response: We regret not having included a discussion of this and thank the reviewer for raising this important issue. Our level of expression of these DREADDs did not differ based on our tissue analyses: There was no significant effect of virus [$F(1,27) = 3.82, p = 0.06$], region [$F(1,27) = 0.331, p = 0.57$], or virus by region interaction [$F(1,27) = 0.594, p = 0.45$].

Additionally, we note that it is once again very rare to add multiple doses of CNO. For example, all of these papers use a single dose of CNO:

<http://www.jneurosci.org/content/35/1/74.long> - 3mg/kg

<http://www.jneurosci.org/content/36/2/385.long> - 3mg/kg

<https://www.ncbi.nlm.nih.gov/pmc/articles/PMC6437658/> - 10mg/kg

<https://www.ncbi.nlm.nih.gov/pmc/articles/PMC6548499/> - 5mg/kg

and we have had success with the 3.0 mg/kg dose in this and other experiments in our lab. Importantly, the pattern of BLA and ACC behavioral effects are quite different, and would be unexpected as plots on the same dose-response curve. Specifically, BLA inhibition results in a leftward shift of waiting times (with *preserved* separation of SNR) whereas ACC inhibition results in a *reduced* separation of SNR and if anything, a slightly rightward shift of waiting times (addressed more fully in response to comment #3 of Reviewer # 3, below). Finally, though it is true that we confirm our DREADDs only in ACC slice recordings, we did also confirm DREADDs by including 3 different control groups to account for nonspecific effects of CNO dose as well as virus spread and exposure, to be thorough. These are more controls and validation than observed in many papers using this technique, in this journal and similar journals.

Most groups do not include all the control groups we include in our study, see below:

<https://www.ncbi.nlm.nih.gov/pmc/articles/PMC4866611/> - *Nature Neuroscience*, only use Lenti-hSyn::hM4Di-CFP virus, not null virus. Single CNO dose 10mg/kg. No DREADD validation.

<https://www.ncbi.nlm.nih.gov/pmc/articles/PMC6202358/> - *Nature Communications*, hM4D and hM3D viruses, but not null viruses. Single CNO dose 1mg/kg.

<https://www.nature.com/articles/npp2015229> - *Neuropsychopharmacology*. Used AAV8-hSyn-GFP-Cre or AAV8- CaMKIIalpha-GFP-Cre with AAV8-hSyn-DIO-hM4D(Gi)-mCherry; no group lacking hM4D(Gi). Single CNO dose 10mg/kg. No DREADD validation.

<https://www.ncbi.nlm.nih.gov/pmc/articles/PMC5645746/> - *Neuropsychopharmacology*. Only active virus. Single CNO dose 0.3mg/kg.

<https://www.ncbi.nlm.nih.gov/pmc/articles/PMC5546988/> - *European Journal of Neuroscience*. Used only hM4Di-containing viruses (but they cite previous papers showing no non-specific effects of CNO). 2 CNO doses 5mg/kg and 10mg/kg. No DREADD validation.

<https://www.ncbi.nlm.nih.gov/pmc/articles/PMC6001626/> - *European Journal of Neuroscience*. Only use active virus. Instead of an inactive virus, they did sham surgeries with no viral infusion. Single CNO dose 3mg/kg.

<https://www.ncbi.nlm.nih.gov/pmc/articles/PMC5800843/> - *eLife*. Only active virus. Single CNO dose 1mg/kg. No DREADD validation.

In addition to reconstructions and quantification, we now have included statistics comparing viral spread in **Supplementary Analysis 8** of the revised the manuscript, and made it clearer we have 4 treatment groups with 3 controls [R1.7].

8. *"I found it very hard to interpret the reversal learning experiment. This is partly because of the issues above, and also the fact that animals were suddenly not allowed to reinitiate trials in this experiment. Were the animals learning two things (not to reinitiate and reverse) at the same time? In the same section in Line 403, it is stated that both LC and HC groups received the same amount of reward in the session. I thought HC group learned faster (and hence more rewards)? I then thought that LC groups have shorter wait time and hence larger number of trials and possibly more rewards over time? These needs further analysis and clarifications."*

Response: We regret the lack of clarity on the reversal learning phase. For the reversal learning phase, we found two pairs of SNR and contrast level in which a given rat demonstrated equivalent accuracy (i.e. matched discrimination accuracy, or d'). Critically, although the performance was equivalent for these two pairs, the rat on average waited longer for one of the pairs. We then randomly chose one of these pairs for each rat for the reversal learning phase. If the rat waited on average less time for one pair over the other, this was designated the low-confidence (LC) condition. If the rat waited on average more time than the other, this was designated the high-confidence (HC) condition. We also confirmed that rats in the HC and LC conditions received the same amount of reward in the session prior to reversal. During reversal learning, the stimulus that signaled the possibility for a trial re-initiation (white square in the middle panel) was removed. This change was introduced to ensure that

the feedback rats received in the reversal learning phase was not a function of waiting time but rather a function of accuracy in learning the reversal policy.

We have now better clarified the approach by which the rats were classified into the HC and LC conditions in the revised manuscript (see [R1.8]).

9. *“Overall, I found it very hard to read the paper. Part of this was due to the fact that the text is buried in between long parentheses dedicated to stat test reports. I counted more than 15 instances in which the stat report is extended over 2-3 lines and often followed immediately by another long stat report. This made it very hard for me to follow the paper in the current format.”*

Response: We agree with the reviewer and regret burying text within such lengthy statistics. In the revision, we initially made tables of equations for the GLM statistics but found them to be far too lengthy (about seven pages) while not helping improve the readability. We finally settled on condensing the statistics reports by presenting the most important information according to similar papers in the journal.

Minor

“Line 183 refers to Figure 2A inset. I do not see an inset there.”

Response: We thank the reviewer for pointing this out. We meant the second axis of the figure in Figure 2A, which is now Figure 3 in the revised manuscript. We have fixed this (see [R1.m1]).

“The analysis in Figure 4C requires further clarification.”

Response: We thank the reviewer for seeking clarification on the analysis in Figure 4C, which now appears in the revised manuscript as Figure 5C.

M-ratio in Figure 5C (in the revised manuscript) is defined as $\text{Meta-}d'/d'$ which varies between 0 and 1, where:

- 0 indicates that the observer’s trial-by-trial confidence rating (waiting time in our experiment) is completely independent of the trial accuracy.
- 1 indicates that the observer perfectly reports the confidence based on the accuracy of the trial. That is, the animal waits longer if the trial is correct and waits shorter if the trial is wrong.

This information can now be found in Methods.

In Figure 5C in the manuscript we have now included a one-way ANOVA, which resulted in a significant difference between groups ($F(2, 267) = 12.65, p = 5.6142 \times 10^{-6}$).

Following BLA inhibition, meta- d'/d' was not significantly different from vehicle (diff(mean)= 0.0509, $p = 0.4430$). However, following ACC inhibition, meta- d'/d' , was significantly lower than vehicle (diff(mean)= -0.2366, $p = 1.6438 \times 10^{-6}$), **Figure 5C**. These results demonstrate that confidence report following ACC inhibition becomes less sensitive to the accuracy of the trial and thus suggest that ACC is involved in the computation of confidence.

This information has been added to the revised manuscript (see [R1.m2]).

"The section looking into the effect of confidence on reversal learning; it is dividing animals to low and high confidence groups. After looking into Methods, I still left unsure how exactly these two groups are defined and matched."

Response: We thank the reviewer for requesting clarification on the grouping of the animals for the reversal learning. We have addressed this point in revision, according to the reviewer's previous comment #8. Because this is a critical detail, we have placed this clarification in the Results instead of the Methods (see [R1.m3]).

Reviewer #2

Reviewer #2 (Remarks to the Author):

"The manuscript describes the first study in rodents to show how confidence, as measured by post-decision waiting times, affects both perceptual inference and learning. Moreover, they show, using DREADDs, that inhibition of anterior cingulate cortex (ACC) and not the amygdala (BLA) disrupts signal-to-noise ratio (SNR) effects on confidence, whereas inhibition of the BLA completely shifts perceptual certainty to slow down reversal learning. This is excellent work and I only have few minor comments:"

Response: We thank the reviewer for the time and effort in the evaluation of our work and the positive feedback. We hope that our responses here and corresponding changes in the revised manuscript address their remaining concerns and questions.

1. *"The finding that high confidence in the stimulus-response rules facilitates reversal learning is intriguing given evidence that lesions of the amygdala and interconnected regions, including ACC, appear to lower evidence thresholds for learning stimulus-outcome reversals and concomitantly strengthen prior beliefs that learned relationships are unstable (Jang et al., 2015). This result could not be attributed to differences in win-stay or lose-shift performance. One explanation, given the results of*

this manuscript, is simply that the monkeys with amygdala and ACC lesions are less confident about their learning. I would be interested in hearing the authors speculate about their results relate to this prior study, and more generally studies that have observed enhanced reversal learning in monkeys and rodents following amygdala lesions (e.g. Stalnaker et al. 2007; Rudebeck et al., 2008). For example, could cueing reversals based on reaching a predetermined criterion represent a high confidence state and are most reversal learning studies biased in this sense? Might a high confidence state also be a way of interpreting the senior author's own finding that monkeys extinguish faster (e.g. Izquierdo and Murray, 2005)?"

Response: We thank the reviewer for these creative and speculative questions that moved beyond the data. A reframing of those earlier studies does seem warranted based on our findings. Indeed, post-training amygdala lesions have been found to facilitate reversal learning (Rudebeck & Murray, JN 2008; Stalnaker, Franz, Singh, & Schoenbaum, Neuron 2007), result in enhanced learning from negative feedback during reversals (Izquierdo et al., JN 2013), and lead to faster instrumental extinction (Izquierdo & Murray, European Journal of Neuroscience 2005). Typically, in these paradigms, animals are extensively pretrained on fully-predictive cue-reward assignments that could lead to a heightened confidence state that, upon amygdala lesion, would result in an inability to update confidence state, enabling quicker learning during the reversal phase.

Given the space constraint, we have done our best to give this intriguing possibility warranted attention. We now have included a brief discussion of it in the revised manuscript (see [R2.1]).

2. *"The distribution of the waiting times doesn't necessarily match the distribution of the reward delivery times, at least over the very short intervals. The waiting time distribution appears Poisson distributed, whereas the reward times follow an exponential distribution. This would seem to imply that the animals are waiting up to a certain minimum threshold. What do the authors think might account for that threshold and is it possible related to some aspect of the task?"*

Response: We thank the reviewer for this observation. We agree that the distribution of the waiting times did not exactly match the distribution of the reward delivery times, yet there is close tracking (see **Supplementary Figure 5**) and the distribution of the former largely depends on the mean, shape, and spread of the latter. As the reviewer suggests, it is likely that rats wait a minimum threshold to inform their choice to either give up or wait longer. However, we observe the waiting time distribution to be skewed given the proximity of expected reward which could indicate a "temporal distortion" as commonly reported in the interval timing literature (Lake, LaBar, & Meck, Neuroscience and Biobehavioral Reviews 2016) near the

median time of reward dispensation, as shown in **Supplementary Figure 5**. When we include all rats in our analyses (**Supplementary Figure 5B**), it appears most rats are waiting past this threshold (i.e. being more patient) as a common strategy.

In the revised manuscript, we now have included a statement about the distribution of waiting times and reward times (see [R2.2]).

3. *“Given that the ACC and BLA manipulations were in different groups of animals it would be helpful to plot the vehicle effects for each group (Fig 3 and S3). I am wondering to what extent in the contrast = 80 condition if averaging the two vehicle groups is possibly masking what is a general effect for the ACC inhibition that resembles that seen in the contrast=40 and contrast=60 conditions. In other words is are the vehicle waiting times for the BLA group especially long in the contrast=80 group.”*

Response: We thank the reviewer for the suggestion of separating the vehicle groups for ACC and BLA manipulations. First, similar to **Figure 3A,B** in the manuscript, we calculated session-based waiting times following ACC inhibition (**Fig. 5A** below) and BLA inhibition (**Fig. 5B** below) and compared their medians with their corresponding vehicle groups (dashed lines). We found no significant differences between targeted brain regions following vehicle on waiting time (GLM: $p = 10^{-16}$; $\beta_{\text{targeted region}} = 0.67, p = 0.83$; **Fig. 5C** below). Furthermore, we separately analyzed trials in which rats responded correctly. As above, we found no significant differences between targeted brain regions following vehicle on waiting time (GLM: $p = 10^{-16}$; $\beta_{\text{targeted region}} = -0.77, p = 0.89$; **Fig. 5D** below). Next, we separately analyzed trials in which rats responded incorrectly to the trial. We similarly found no significant effect of targeted brain region following vehicle on waiting time (GLM: $p = 1.45 \times 10^{-228}$; $\beta_{\text{targeted region}} = 2.12, p = 0.55$; **Fig. 5E** below).

Figure 5. No evidence for difference in waiting time following vehicle administration in either ACC or BLA. (A) Waiting time increases following ACC inhibition. Plotted are the distributions of the waiting time separately for each SNR, following ACC inhibition. Solid lines show the median of each distribution. Dashed lines show the median of the same condition but following vehicle administration. (B) Waiting time decreases following BLA inhibition. Same as in panel A but for sessions following inhibition of BLA. (C) Waiting time for all trials together is similar for the rats with ACC as the targeted brain region following vehicle administration and the rats with BLA as targeted brain region following vehicle administration. Plotted is the waiting time for all trials as a function of SNR following vehicle administration for rats with ACC as the targeted brain region (blue dashed line) and BLA as the targeted brain region (orange dashed line), inhibition of ACC (blue), and inhibition of BLA (orange). (D-E) Same as in panel C but only on trials in which a correct (D) or incorrect (E) response was made. Error bars show the S.E.M. over sessions (typically smaller than the symbols).

We also separately analyzed stimuli based on the contrast and found no significant differences in waiting time following vehicle administration in either ACC or BLA (GLM contrast = 40: $p = 1.27 \times 10^{-136}$; $\beta_{\text{targeted region}} = -1.461, p = 0.26$; Fig. 6A below; GLM contrast = 60: $p = 1.3 \times 10^{-260}$; $\beta_{\text{targeted region}} = -0.7, p = 0.63$; Fig. 6B below; GLM contrast = 80: $p = 10^{-16}$; $\beta_{\text{targeted region}} = -2.66, p = 0.11$; Fig. 6C below). Together, these results show that averaging the two vehicle groups is not masking a general effect of ACC inhibition that resembles what that was seen for contrast=40 and contrast=60 conditions. Furthermore, the vehicle waiting times were not significantly different between targeted regions.

Figure 6. No difference in waiting times for different contrast levels following vehicle administration in either ACC or BLA. (A) Waiting time is similar following vehicle administration in ACC or BLA. Plotted is the waiting time for all trials as a function of SNR and for contrast of 40 following vehicle administration for rats with ACC as the targeted brain region (blue dashed line) and BLA as the targeted brain region (orange dashed line), inhibition of ACC (blue), and inhibition of BLA (orange). (B-C) Similar to panel A but for contrast of 60 (B) and 80 (C).

In the revised manuscript, we have now included the important point about no evidence for difference following vehicle administration in either ACC or BLA (see [R2.3]). In addition, we have added **Figs. 5** and **6** above as new **Supplementary Figures 5** and **6** in the revised manuscript.

4. *“With respect to the odd effects of contrast on the waiting times, is this possibly due to individual differences in perception among the rats tested? Instead of averaging across the three contrast levels, and effectively ignoring them, would it be possible to try and determine high and low contrast possible using d' as a sorting metric? I’m curious because the results look very similar between the 40 and 60 conditions, and I am wondering if the non-linear effects in the 80 condition represent hitting some perceptual threshold in some animals and not others.”*

Response: We thank the reviewer for raising the concern of perceptual threshold. In order to address this concern, we further analyzed the effect of contrast and SNR on accuracy. First, we separated the trials by contrast and computed accuracy as a function of SNR. We observed that accuracy increases as SNR increases for each of the contrast levels (**Fig. 7A** below; GLM contrast=40: $p = 1.14 \times 10^{-27}$; $\beta_{SNR} = 0.108$, $p = 1 \times 10^{-3}$; GLM contrast=60: $p = 6.07 \times 10^{-32}$; $\beta_{SNR} = 0.157$, $p = 2.08 \times 10^{-8}$; GLM contrast=80: $p = 1.19 \times 10^{-11}$; $\beta_{SNR} = 0.087$, $p = 2 \times 10^{-3}$). Observing the effect for each contrast, we also calculated accuracy as a function of SNR across all contrast values. Similarly, we found that accuracy increases with SNR on accuracy (**Fig. 7B** below; GLM: $p = 5.01 \times 10^{-70}$; $\beta_{SNR} = 0.118$, $p = 1.23 \times 10^{-11}$).

These results illustrate that for easier stimuli (higher SNR), performance is better compared to the more difficult stimuli (lower SNR).

Figure 7. Accuracy increases for easier stimuli. (A) Accuracy is plotted as a function of SNR for different values of stimulus contrast. The squares and error bars show the mean and S.E.M. The solid lines show the fitted psychometric curve over the observed data. Different values of contrast are indicated in the legend and correspond to least discriminable (contrast = 40), moderately discriminable (contrast = 60), and most discriminable (contrast = 80). (B) Plots shows the average accuracy as a function of SNR across all contrast levels.

Next, we looked at d' as a function of contrast and SNR. We found that similar to accuracy, both contrast and SNR have significant effects on d' (GLM: $F(3,131) = 61.2, p = 8.64 \times 10^{-25}$, adjusted $R^2 = 0.574$; $\beta_{SNR} = 0.48, p = 1.8 \times 10^{-4}$; $\beta_{contrast} = 0.02, p = 1.7 \times 10^{-4}$; Fig. 8 below). Furthermore, there was no significant effect of contrast on d' in comparing contrast of 60 and 80 (GLM: $F(3,86) = 28.5, p = 6.81 \times 10^{-13}$, adjusted $R^2 = 0.48$; $\beta_{contrast} = 0.02, p = 0.12 (> p^* = 0.05/3)$).

Overall, these results show that the accuracy for contrast of 60 and 80 are equal and therefore, it is possible that animals reach a perceptual threshold. Importantly, however, the waiting times for contrast of 80 are significantly higher than those for contrast of 60.

We have mentioned and discussed these results in the revised manuscript to address the reviewer's concern (see [R2.4]). We also have added Fig. 7 above and Fig. 8 below as new Supplementary Figure 4 to the revised manuscript.

Figure 8. d' increases with both SNR and contrast. Plot shows d' as a function of SNR for different values of contrast. The squares show the average d' and error bars show S.E.M. Different values of contrast are indicated in the legend and correspond to least discriminable (contrast = 40), moderately discriminable (contrast = 60), and most discriminable (contrast = 80).

5. *“I would be interested to see the correlation for the reaction and waiting times broken out by correct and incorrect trials. Also, are the RTs plotted in Figure 2 pooled across all of the animals? Why not simply plot a histogram of the within session correlations as that is reported in the results.”*

Response: We thank the reviewer for asking about the correlation of waiting time and reaction time for correct and incorrect trials. To address these questions, we first calculated the trial-by-trial correlation of waiting time and reaction time for correct and incorrect trials (**Fig. 9** below). Next, we calculated the correlation of waiting time and reaction time in each session and compared the distribution of the session-based correlations for correct and incorrect responses (**Fig. 10** below). We observed that across sessions, the correlation between waiting time and reaction time is significantly different and associated more negatively for correct trials relative to incorrect trials (mean(diff) = -0.074 ; paired-sample t-test; $t(146) = -4.53, p = 1.2 \times 10^{-5}$).

Finally, considering the reviewer’s question about RTs in **Figure 2**, we confirm that the RTs are pooled across all the animals. We have also performed analyses based on session-by-session normalization of RTs per request of the Reviewer #1 (comment #5) which yields to similar conclusions.

We have now incorporated **Figs. 9** and **10** below as one **Supplementary Figure 6** in the revised manuscript (see [R2.5]).

Figure 9. Waiting time before re-initiation of a new trial is negatively correlated with reaction time to make a choice. Waiting time is plotted as a function of the reaction time for all rats in correct trials (A) and incorrect trials (B). Each data point is a trial in a session following vehicle administration.

Figure 10. The correlation between waiting time and reaction time is significantly stronger in correct trials than in incorrect trials. Plotted are distribution of the session-based correlation between waiting time and reaction time for all trials (black), correct trials (green), and incorrect trials (blue). Each solid line shows the median of each distribution. (*) indicates $p < 0.05$.

6. "The double-y plots in Figures 2 are quite confusing and not well explained in either the figure legend or text. What elements relate to each of the y-axes? Perhaps it would be simpler to just break out the plots individually as is done in Figure 3?"

Response: We regret the lack of clarity and thank the reviewer for asking about the missing legend for the double-axis panels. In panel A, the y-axis on the right shows the probability of a correct response. Solid circles indicate the calculated probability of correct responses for each bin of the waiting time distribution, for each SNR. The dashed lines show the regression line of the probability of correct responses for bins in each SNR. The y-axis in panel B is similar to panel A, but for the reaction time.

We now have added the missing legend to the revised manuscript (see [R2.6]).

7. "In line 182 I believe you are referring to the inset in Figure 2D and not 2A, where there is no insert"

Response: We thank the reviewer for pointing out the miscommunication here. We meant the second y-axis of Figure 2A. We now have revised it in the manuscript (see [R2.7]).

Reviewer #3

Reviewer #3 (Remarks to the Author):

"Stolyarova, Rakhshan et al. used chemogenetic inactivation of the ACC and BLA during a temporal-wagering perceptual decision-making task designed to measure decision confidence in rats. In this task, increased waiting times for variable-delay rewards are considered to be a measure of decision confidence. The authors report that BLA inactivation decreased waiting times by similar amounts regardless of stimulus strength and SNR, while ACC inactivation increased waiting times in a fashion that rendered them insensitive to the strength of evidence. Moreover, they show that in rats higher confidence can lead to faster learning of reversed stimulus-reward associations, and that either BLA or ACC inactivation abolishes the effect of higher confidence, albeit by changing behavioral strategies in different ways. The authors conclude that only ACC activity is required to compute confidence on a trial-by-trial basis, but that both ACC and BLA are required for learning that relies on decision confidence.

The questions posed by this study are interesting and the results are novel and of interest to the community. Moreover, the manuscript is clearly written. However, my enthusiasm is decreased by technical and interpretational issues that I believe should be addressed before I can recommend the manuscript for publication."

Response: We thank the reviewer for the time and effort in the evaluation of our work and the positive and critical feedback. We hope that our responses here and corresponding changes in the revised manuscript address their remaining concerns and questions.

Major issues:

1. *“Perhaps I missed it, but I could not find a crucial inactivation control, namely CNO delivery in animals not expressing DREADDs (e.g. injected with the EGFP construct). While the other controls included in the study are important, they do not address the possibility that the effects of inactivation are not (or at least not exclusively) mediated by silencing of the neurons expressing DREADDs, but by effects of clozapine on its target receptors. CNO has been shown to cross the blood-brain barrier with very low efficiency, such that the effects of delivering it systemically are mostly due to its byproduct clozapine entering the brain and both activating DREADDs and other target receptors (Gomez et al, Science, 2017; Mahler and Aston-Jones, Neuropsychopharmacology, 2018). The authors may argue that the effects are different in BLA and ACC animals, but that could still be due to the interaction of DREADDs activation and that of other receptors that clozapine binds. The slice recordings also do not address this possibility since the blood-brain barrier is not a variable there.”*

Response: We thank the reviewer for pointing out concerns about the appropriate DREADDs controls. We completely agree this was essential and indeed we had 3 different control groups for which we reported the results in the **Supplementary Analysis 7**: 1) eGFP-CNO, 2) hM4Di DREADD-VEHICLE, and 3) eGFP-VEHICLE. We did not find any difference between any of these control groups and thus used the results from the DREADDs-VEHICLE vehicle against the DREADDs -CNO group.

We have also included **Figs. 5 and 6** above that plot the results for vehicle groups (requested by Reviewer # 2) as new **Supplementary Figures 5 and 6** to the revised manuscript in order to show more clearly that there were no significant differences in waiting time or reaction time following vehicle administration in either ACC or BLA. We also have made results on comparison of the three control conditions more prominent (see [R3.1] in the revised manuscript).

2. *“Some puzzling aspects of task performance are not explained in the manuscript, specifically:”*

2.1. *“It seems from figure S1 that performance does not increase monotonically with contrast. For example, for SNR = 4, performance is higher for 60% than 80% contrast. It would be helpful to show a panel in this figure of performance (d') as a function of contrast and SNR, as it is hard to extract that*

information from panel B. Related to this, was the luminance of the screens gamma-corrected?"

Response: We thank the reviewer for raising the concerns about the differential effect between contrast levels 60 and 80 for high SNR stimuli (SNR = 4). A similar point was also mentioned by Reviewers #1 and # 2. As is shown in **Fig. 11** below, both contrast and SNR have significant effects on d' (GLM: $F(3,131) = 61.2, p = 8.64 \times 10^{-25}$, adjusted $R^2 = 0.574$; $\beta_{SNR} = 0.48, p = 1.8 \times 10^{-4}$; $\beta_{contrast} = 0.02, p = 1.7 \times 10^{-4}$). However, there was no significant effect of contrast on d' in comparing contrast of 60 and 80 (GLM: $F(3,86) = 28.5, p = 6.81 \times 10^{-13}$, adjusted $R^2 = 0.48$; $\beta_{contrast} = 0.02, p = 0.12 (> p^* = 0.05/3)$).

With respect to gamma correction: Screen luminance was *not* gamma corrected. The stimuli were, however, designed to have the same average luminance based on pixel values, using a photo/lux meter. This presents the limitation that the contrast values themselves will not be informative, but because the goal was to titrate to a certain performance level, the reported data showed that the goal was achieved, regardless of what contrast values were used to achieve this. In other words, we were not looking for contrast sensitivity thresholds which would require a linear luminance/contrast profile, but the important factor was that performance was above threshold but not at ceiling, which it was.

In the revised manuscript, we now have included **Fig. 10** below as part of a new **Supplementary Figure 4** and discussed its results (see [R3.2]).

Figure 11. d' increases with both SNR and contrast. Plot shows d' as a function of SNR for different values of contrast. The squares show the average d' and error bars show S.E.M. Different values of contrast are indicated in the legend and correspond to least discriminable (contrast = 40), moderately discriminable (contrast = 60), and most discriminable (contrast = 80).

2.2. *“In Lak et al, 2014, waiting times in error trials decrease with stimulus evidence strength, which is predicted by their normative model. However, in the present manuscript, the opposite trend is seen in the data. The authors discuss that this could be due to differences in sensory modality and food vs water reward, but it is also possible that the present task structure imposes different normative constraints. Have the authors tried fitting Lak et al’s model to their behavioral data, or can they provide alternative models?”*

Response: We thank the reviewer for raising the concern about the observed increase in waiting times as a function of stimulus strength in error trials. A similar concern was also raised by Reviewer # 1 (comment #2, see [R1.2]). As the reviewer points out, several studies including the work by Lak and colleagues have observed an opposing pattern of confidence readout with stimulus strength (referred to as divergence signature of the confidence). In contrast, we find that waiting time as a proxy for confidence decreases in error trials as the stimulus becomes stronger (easier to discriminate). Nonetheless, there are recent experimental and theoretical studies showing that confidence can increase on easier trials even for incorrect trials and thus, suggest that the divergence signature of confidence is not necessarily the *defining* signature of confidence (Kiani, Corthell, & Shadlen, Neuron 2014; Rausch, Hellmann, & Zehetleitner, Attention, Perception, and Psychophysics 2018; van den Berg et al., eLife 2016).

For example, Kiani and colleagues (2014) showed that in a random-dot motion discrimination task in which subjects report choice and certainty simultaneously, certainty grows with stimulus strength for both correct and error responses (see their Figure 4). In order to explain the violation of the divergence signature of confidence based on normative models, they used a modified version of a bounded accumulation model. In this model, confidence in choosing an option is a function of both the amount of accumulated evidence supporting that option and the accumulation time. In another study, van den Berg and colleagues (2016) used a motion discrimination task to show that when the subjects are more confident, they tend to be more accurate. Subjects were more confident on correct compared with error trials but more importantly, the subjects were more confident with stronger motion coherence even on error trials. The authors used a bounded accumulation mechanism to explain how the time to decision influences confidence and why confidence in error trials increases with stronger motion coherence. Finally, Rausch et al. (2018) used two sets of experiments in which the subjects reported the orientation of a grating while the stimulus was masked with noise. They observed that confidence in both correct and error trials increase with the stimulus strength. They further proposed a probabilistic normative model (weighted evidence and visibility model) to account for the observed pattern of confidence report.

Importantly, there are also recent studies based on Bayesian framework that argue the divergence signature of confidence disappears under different conditions of stimuli, noise, and task structure (Adler & Ma, Neural computation 2018; Rausch & Zehetleitner, bioRxiv 2018). For example, Adler and Ma (2018) showed that the divergence signature of Bayesian confidence is present if category-conditioned stimulus distributions are uniform, independent of high- and low-noise regimes. Critically, they show that if the stimulus distributions are Gaussian and measurement noise is relatively low compared to the stimulus distribution width, then the divergence signature is not expected.

Similarly, by extending Bayesian inference to include evidence about stimulus strength, Rausch & Zehetleitner (2018) illustrated that the divergence signature is a special case of confidence signatures when the evidence about stimulus strength is omitted from the calculation of objective confidence (i.e., only stimulus identity is available). More specifically, they found that previous studies that observed a violation of the divergence signature of confidence used tasks in which the stimulus was composed of a feature that assessed the correct response along with an orthogonal manipulation of stimulus strength. In fact, this is the case in our experiment in which we assessed accuracy in perceptual judgment by changing SNR and contrast of the stimuli as two orthogonal manipulations.

Together, recent studies reviewed above clearly demonstrate that orthogonal components of the stimuli and the structure of the confidence report (e.g. simultaneous choice and confidence in choice vs. a two-step task) can change what has been considered as “the” signature of confidence. We have now revised the manuscript to better discuss the interpretation of our results that violate of the so-called statistical signature of confidence (see [R3.3]).

3. *“The authors report that ACC, but not BLA, inactivation abolishes sensory-evidence dependence of wait times and conclude that this is a signature of ACC’s role in confidence computations. However, the waiting times as increased across all SNRs with ACC inactivation, mimicking increased decision confidence. It seems to me that the most likely effect of removing confidence computations would be to decrease waiting times for high-confidence trials (like the effects of OFC inactivation in Lak et al). This is true both from a normative standpoint (assuming rats try to maximize reward rate), and from the empirical observation that rats tend to be impulsive. Applying similar logic to the authors’ explanation of the BLA results, an alternative explanation for these findings is that inhibiting ACC has a non-specific effect, e.g. decreases impulsivity, and that the loss of dependence on SNR is an artifact coming from a ceiling effect, i.e. that waiting times cannot be longer because of task design.”*

“The authors discuss that these ACC findings are compatible with a role for this structure in gain-modulating uncertainty signals, but this claim is incompatible with the effects of microstimulating

sensory areas on decision confidence (Fetsch et al, Neuron, 2014), and with the known effects of stimulating ACC on visual responses (Zhang et al, Science, 2014)."

Response: We thank the reviewer for this important comment that cuts straight into the motivation of this experiment and its interpretation. We think the finding that ACC inhibition results in a significantly reduced separation of waiting times by SNR is in fact more detailed evidence that rats are not using the sensory information to guide their waiting times. Critically, the effect of ACC inhibition is not a uniform increase in waiting time for all SNR; the z-scored waiting time for SNR=4 in ACC inhibition is even decreased compared to vehicle (two sided t-test: $t(213) = 12.41, p = 5.4 \times 10^{-27}$). This suggests that ACC inhibition removes the sensitivity to SNR and does not simply increase waiting time. Additionally, the distributions of waiting times are quite symmetric in all conditions, and their means and medians are very far from the 40 sec maximum waiting time. Interestingly, the animals wait more than 30 seconds in ~2% of the trials even though receiving reward was very unlikely for such long delays (see **Supplementary Figure 7**).

As the reviewer points out, our pattern of results is different than those reported by Lak et al. (muscimol inactivation of OFC reduces waiting time on the most certain trials). Both our results and those of Lak et al., however, could be interpreted as a reduced variance in waiting time that indicates rats may be using an average (~15 sec in our case) instead of sensory evidence (see similarity with Figure 4A in Lak et al., Neuron 2014). Importantly, the reaction times in our hands did not exhibit this pattern, suggesting that it is not simply a motor timing effect, but instead a confidence readout.

As for the possibility that our effects are consistent with decreased impulsivity following ACC inhibition, the prior literature in rats does not support this interpretation. ACC inactivation actually *increases* impulsivity (Hosking, Cocker, & Winstanley, Neuropsychopharmacology 2014). In fact, there is much more evidence that OFC inhibition (pharmacological inactivation) results in a wager sensitive phenotype (more conservative choices) (Barrus, Hosking, Cocker, & Winstanley, Neuroscience 2017) that would be consistent with the findings by Lak and colleagues. Relatedly, there is strong support for lesions and/or pharmacological inactivation of BLA resulting in a delay averse (impulsive) phenotype and we take the overall leftward shift in their willingness to wait to be consistent with those findings. Importantly, the separation between the SNRs in waiting times was preserved/intact following BLA inactivation.

Finally, we thank the reviewer for pointing us to the very important papers by Fetch et al. 2014 and Zhang et al. 2014, which are now discussed in the revised manuscript (see [R3.4]). We note that our finding on gain modulation of uncertainty signals by ACC is not incompatible with an involvement of cingulate on sensory modulation and contributions of

sensory signals to confidence computations. More specifically, ACC inhibition could reduce center-surround modulation of visual responses which, in turn, negatively impact confidence computations. Nevertheless, we would like to be cautious in comparing results across different techniques used in those papers and our (microstimulation in monkeys, optogenetic activation in mice, and DREADDs inhibition in rats). Both microstimulation and optogenetic activation are more acute manipulations than a chemogenetic approach used here.

We have added this important possibility raised by the reviewer and a comparison with Lak et al's results in the Discussion. In addition, we have included a brief discussion of these analyses mentioned above in the revised manuscript (see [R3.4]).

Along the lines of my point 2.2, fitting a normative model may help disambiguate these possibilities, as would repeating the experiments with longer reward delays."

Response: We thank the reviewer for this suggestion. We hope that our responses for the previous comment and comment #2 (existing normative models of confidence) have convinced the reviewer that our observations are backed by most recent literature. Considering that recent Bayesian/normative models can reproduce our observed pattern of confidence as a function of stimulus strength (Kiani, Corthell, & Shadlen, Neuron 2014; Adler & Ma, Neural computation 2018; Rausch & Zehetleitner, bioRxiv 2018; van den Berg et al., eLife 2016), we do not think that fitting our data based on a normative model would provide additional insight. Nonetheless, we plan to construct a multi-area mechanistic model to account for our inactivation experiments, but this is beyond the scope of this study.

Minor

"Figure 1C. Maybe it's an artifact of my pdf, but the histology figures are hard to see. Can the authors provide higher-resolution figures? Also, please add scale bars. Finally, can the authors add quantification of the location of the injection sites, as well as injection spreads? This is important for interpretation of the results, since the injections could encompass neighboring structures."

Response: We thank the reviewer for asking about this important validation and quantification. Per this comment and that of Reviewer # 1, we now have added additional higher-resolution photomicrographs and reconstructions of the DREADDs expression to the revised manuscript (**Supplementary Analysis 8**) and discussed its relevance to our behavioral results (see **Fig. 4** above). This appears in the revised manuscript as new **Figure 3**(see [R3.m1]).

"A few statements are not backed by references, for example lines 175 and 210. Please add."

Response: We thank the reviewer for pointing out these omitted references. This information has been added to the revised manuscript (see [R3.m2]).

“The finding that ACC inhibition does not affect discrimination performance (Fig 4) seems surprising, given many previous findings in multiple sensory modalities that mPFC/ACC/M2 activity manipulations do affect perceptual decision-making performance in rodents (e.g. Rodgers and DeWeese, Neuron, 2014; Zhang et al., Science, 2014; Pinto and Dan, Neuron, 2015; Hanks et al., Nature, 2015; Goard et al., eLIFE, 2016; Huda et al., bioRxiv, 2018). Can the authors add a discussion about potential reasons?”

Response: We thank the reviewer for suggesting this important discussion. On reading the suggested citations, most of the references describe neural correlate techniques (recording/imaging), not causal manipulations. Thus, the signals may be related to perceptual decision making, but may not be required. There were exceptions; Hanks et al. 2015 examined the causal role of FOF by optogenetically silencing it during different time points in perceptual decision making. This group found that FOF was required for *committing* to a “categorical choice at the end of the evidence accumulation process.” Similarly, Goard et al. 2016 featured both recording and optogenetic techniques and found that frontal motor was important in *maintaining* “the choice in memory prior to execution.” These are all nearby regions to ACC, and we have included a statement about the possibility that more temporally-precise manipulations may yield different findings (see [R3.m3] in the revised manuscript).

“Fig 4C. The more rigorous statistical approach here would be to carry out an ANOVA first, followed by post-hoc tests.”

Response: We thank the reviewer for suggesting this change to the statistics.

A one-way ANOVA resulted in a significant difference between groups ($F(2, 267) = 12.65, p = 5.6142 \times 10^{-6}$). Following BLA inhibition, meta- d'/d' was not significantly different relative to vehicle (diff(mean)= 0.0509, $p = 0.4430$). However, following ACC inhibition, meta- d'/d' , was significantly lower compared to vehicle (diff(mean)= -0.2366, $p = 1.6438 \times 10^{-6}$). These results demonstrate that confidence report following ACC inhibition becomes less sensitive to the accuracy of the trial and consequently, that ACC is involved in the computation of confidence.

We now have added the aforementioned analyses to the revised manuscript (see [R3.m4]).

“The results in Figure 5A are very interesting and the control analyses presented in the paragraph starting in line 392 are important. To help readers, I would suggest adding panels to Figure 5 or a supplemental Figure showing the results of these analyses, rather than just reporting them in the text.”

Response: We thank the reviewer for the instructive suggestion to show the control analysis through figures. We have now included a **Supplementary Figure 12** that points to each piece of the statistics in that paragraph. Following is the revised part in the manuscript:

We performed several analyses to ensure that the only difference between HC and LC conditions was the confidence reported via waiting time. First, we found that d' for HC and LC conditions were not significantly different for each of the stimuli that was administered after reversal, i.e., the contrast-SNR pairs that were chosen for reversal were not associated with different d' before reversal (Stepwise GLM: $F(3,11) = 2.46, p = 0.11$, adjusted $R^2 = 0.238$; $\beta_{\text{confidence}} = 0.43, p = 0.32$; **Fig. 12A** below). Second, d' for HC and LC conditions were not significantly different across all contrast-SNR pairs (GLM: $p = 8.2 \times 10^{-16}$; $\beta_{\text{confidence}} = -0.14, p = 0.92$; **Fig. 12B** below). Third, metacognitive efficiency (meta- d'/d') across HC and LC conditions was not significantly different for the specific contrast-SNR pair that was used after reversal (Stepwise GLM: $p = 0.44$; $\beta_{\text{confidence}} = -0.1, p = 0.44$; **Fig. 12C** below). Fourth, rats in both HC and LC conditions acquired equal amount of reward in the no-injection control session for the specific pair of contrast and SNR values (Stepwise GLM: $p = 0.01$; $\beta_{\text{confidence}} = -0.2, p = 0.08$; **Fig. 12D** below). Finally, we found that HC and LC conditions were different in waiting time, reflecting confidence, for the specific contrast-SNR pair that was used after reversal (Stepwise GLM: $p = 2.7 \times 10^{-31}$; $\beta_{\text{confidence}} = -28.3, p = 2.81 \times 10^{-9}$; **Fig. 12E** below). Together, these results illustrate that the only difference between HC and LC conditions was the confidence.

In the revised manuscript, we have now added **Fig. 12** below as **Supplementary Figure 12** and discussed its results (see [R3.m5]).

Figure 12. The only difference between HC and LC conditions was the waiting time in the session prior to reversal experiment. (A) d' for HC and LC conditions were not significantly different for each of the stimuli that was administered after reversal. Plotted is the d' for HC (yellow) and LC (orange) conditions for the contrast-SNR pairs that were chosen for reversal. Error bars show S.E.M. **(B)** d' for HC and LC conditions were not significantly different across all contrast-SNR pairs. Plotted is the d' for HC (yellow) and LC (orange) conditions for all the contrast-SNR pairs. **(C)** Metacognitive efficiency ($meta-d'/d'$) across HC and LC conditions was not significantly different for the specific contrast-SNR pair that was used after reversal. Plotted is the $meta-d'/d'$ for HC (yellow) and LC (orange) conditions for each of the stimuli that was administered after reversal. **(D)** Both HC and LC conditions acquired equal amount of reward in the no-injection control sessions for the specific pair of contrast and SNR values administered after reversal. Plotted are the distribution of the session-based average rewarded trials for HC (yellow) and LC (orange) conditions. **(E)** HC and LC conditions were different in waiting time for the specific contrast-SNR pair that was used after reversal. Plotted are the distribution of the session-based average waiting time for HC (yellow) and LC (orange) conditions.

“Figures 5B, 6B, 6D. Please add error bars / confidence intervals. Also, did the authors correct for multiple comparisons when calculating significance for these tests?”

Response: We thank the reviewer for asking about the error bars. The reason that those specific figures do not have error bars is because of the statistical approach we have used. In those figures, the comparisons are between 3 or 4 rats (because the initial group sizes were 7-8 and we divided the groups further into high confidence and low confidence conditions). Therefore, the usual parametric statistics are inappropriate. In order to increase the statistical power, after consultation with a statistician, we concatenated the choices of all rats and then used a permutation of the resulting array to test the probability that the actual value would occur over 10,000 permutations of that array. Therefore, since the permutation was conducted

on the concatenated array, there is actually one set of data and we could not calculate error bars for that array. Finally, in order to calculate the significance, we have used Bonferroni correction of p-values.

This procedure and correction appear in the Methods section (see [R3.m6]).

“Figure 6A. This is related to my major point #3 — is this incompatible with the results in Figure 3? In other words, if ACC inactivation mimics the effect of increased confidence, wouldn't one expect faster learning? Maybe this is a naive interpretation but I would recommend expanding the discussion on this point (beyond line 659-660).”

Response: We thank the reviewer for pointing out this concern. However, as explained in more detail in response to point # 3 above, we respectfully disagree with the premise that ACC inactivation mimics increased confidence. The effect of ACC inhibition on individual values of SNR is not uniform (there is not a uniform increase in waiting time for all SNR). In fact, what we conclude based on the collective data is that ACC inactivation reduces the ability of SNR to influence waiting time (confidence reports).

We have tried to make this more clear throughout the manuscript (see [R3.m7]).

“Methods, line 850. Can the authors justify why these trials were excluded?”

Response: We thank the reviewer for asking about exclusion criterion. We have used the exclusion criteria that removes the trials in which the z-scored reaction time has been more than 3. This exclusion criteria considered because there are trials in which the rat starts moving around and ignore the stimuli on the screen for a while, after demonstrating its intention to start the trial. That is why we have removed those trials by considering the aforementioned exclusion criteria.

We have now clarified this point in the revised manuscript (see [R3.m8]).

“Supplementary analyses (page 33). I suggest showing these as figures instead of text to help the reader.”

Response: We thank the reviewer for this comment which would increase the readability of the manuscript. We have revised the following in the manuscript and added **Fig. 13** below as **Supplementary Figure 9** (see [R3.m9]).

We first observed that the type of virus (null vs. active) had no significant main effect or interaction with trial type (correct vs. incorrect) and/or SNR values on waiting time (GLM:

$p = 6.12 \times 10^{-57}$; $\beta_{\text{virus type}} = 1.61, p = 0.3$; $\beta_{\text{virus type} \times \text{trial type}} = -0.16, p = 0.94$; $\beta_{\text{virus type} \times \text{ratio}} = -0.008, p = 0.98$; $\beta_{\text{virus type} \times \text{trial type} \times \text{ratio}} = -0.35, p = 0.62$; **Fig. 13A** below). Similarly, we did not find a significant effect of virus type in either main effect or interactions with trial type and SNR on reaction time ($p = 9.02 \times 10^{-77}$; $\beta_{\text{virus type}} = 0.09, p = 0.23$; $\beta_{\text{virus type} \times \text{trial type}} = -0.13, p = 0.25$; $\beta_{\text{virus type} \times \text{ratio}} = -0.03, p = 0.15$; $\beta_{\text{virus type} \times \text{trial type} \times \text{ratio}} = 0.03, p = 0.44$; **Fig. 13B** below).

Next, we analyzed the effect of virus type on performance measures of probability of correct response, d' , and $\text{meta-}d'/d'$. We observed no significant effect of virus type or interaction with SNR on probability of correct response (GLM: $p = 2.16 \times 10^{-23}$; $\beta_{\text{virus type}} = -0.03, p = 0.15$; $\beta_{\text{virus type} \times \text{ratio}} = 0.01, p = 0.16$; **Fig. 13C** below). Furthermore, we observed that the type of virus had no significant effect or interaction with SNR values on d' (GLM: $p = 2.13 \times 10^{-45}$; $\beta_{\text{virus type}} = -0.35, p = 0.6$; $\beta_{\text{virus type} \times \text{ratio}} = 0.15, p = 0.48$; **Fig. 13D** below). Finally, we observed no significant effect of virus type or interaction with ratio on $\text{meta-}d'/d'$ (GLM: $p = 0.005$; $\beta_{\text{virus type}} = -1.01, p = 0.27$; $\beta_{\text{virus type} \times \text{ratio}} = 0.12, p = 0.68$; **Fig. 13E** below). These results show that the observed impairments through CNO administration was specific to the active (DREADDs) virus we used in the experiment.

Figure 13. Absence of non-specific effects of virus exposure. (A) the type of virus (null vs. active) had no significant main effect on waiting time. Plotted are the distribution of the session-based average waiting time with null (EGFP) and active (DREADDs) virus following vehicle administration in orange and yellow, respectively. The solid lines show the mean of the distributions. (B) Similar to panel A but for reaction time. Conventions are similar to panel A. (C) Similar to panel A but for the probability of correct response. Conventions are similar to panel A. (D) The type of virus had no significant effect on d' . Plotted is the d' for active (yellow) and null (orange) virus. Error bars show S.E.M. (E) Similar to panel D but for $\text{meta-}d'/d'$. Conventions are similar to panel D.

Reviewers' Comments:

Reviewer #1:

Remarks to the Author:

The manuscript is substantially improved, and I have no further comment.

Reviewer #2:

Remarks to the Author:

The authors have thoroughly addressed all of my concerns and I enthusiastically recommend the manuscript for publication.

Reviewer #3:

Remarks to the Author:

The authors have done a good job addressing the reviewers' comments and the manuscript is much improved. Most of my concerns have been satisfactorily addressed but one important issue still remains. The authors mention in the results, methods and their reply to me that they do have an eGFP-CNO control. However, I still cannot find it in Supplementary Analyses 5 or 7, or at least it is unclear to me which statistics in those paragraphs refer to said control. Figure S9 speaks only to the specificity of the viral constructs, since it only presents data from vehicle injections in animals expressing either DREADDs or EGFP. This does not address the concern that metabolic byproducts of CNO (clozapine) act in a DREADDs-independent fashion, through neuromodulation rather than direct inhibition. I think it is crucial to add the eGFP-CNO data in figure format.

Response to reviewers' comments, and summary of changes made in response to their comments.

Title: Dissociable roles for Anterior Cingulate Cortex and Basolateral Amygdala in Decision Confidence and Learning under Uncertainty

Authors: Stolyarova, Rakhshan, Hart, O'Dell, Peters, Lau, Soltani, and Izquierdo

We are thankful to the reviewers for their careful reading of our revised manuscript and for their positive feedback. We have now included missing analyses to address the last concern regarding the possible effect of metabolic byproducts of CNO (clozapine) in a DREADDs-independent manner. The corresponding changes have been clearly marked (in blue) in the revised manuscript.

Reviewer comments

Reviewer #1

Reviewer #1 (Remarks to the Author):

"The manuscript is substantially improved, and I have no further comment. "

Response: We thank the reviewer for the time and effort in the evaluation of our work. We are happy that our responses in the revision addressed the reviewer's concerns.

Reviewer #2

Reviewer #2 (Remarks to the Author):

"The authors have thoroughly addressed all of my concerns and I enthusiastically recommend the manuscript for publication."

Response: We thank the reviewer for the time and effort in the evaluation of our work and the positive feedback. We are happy that our responses in the revision fully addressed the reviewer's concerns.

Reviewer #3

Reviewer #3 (Remarks to the Author):

“The authors have done a good job addressing the reviewers’ comments and the manuscript is much improved. Most of my concerns have been satisfactorily addressed but one important issue still remains.”

Response: We thank the reviewer for the time and effort in the evaluation of our work and the positive feedback. We hope that our final response below addresses their remaining concern.

“The authors mention in the results, methods and their reply to me that they do have an eGFP-CNO control. However, I still cannot find it in Supplementary Analyses 5 or 7, or at least it is unclear to me which statistics in those paragraphs refer to said control. Figure S9 speaks only to the specificity of the viral constructs, since it only presents data from vehicle injections in animals expressing either DREADDs or EGFP. This does not address the concern that metabolic byproducts of CNO (clozapine) act in a DREADDs-independent fashion, through neuromodulation rather than direct inhibition. I think it is crucial to add the eGFP-CNO data in figure format.”

Response: We thank the reviewer for pointing out the importance of illustrating that CNO effects depend on DREADDs. We agree that a comparison of eGFP-vehicle and eGFP-CNO is crucial for our claims and simply missed reporting it previously. The comparisons between eGFP-vehicle and eGFP-CNO rats have now been reported in **Supplementary Analysis 10**, some of which are also plotted in **Supplementary Figure 14**.

We first observed that there was no significant effect of drug administration type (vehicle vs. CNO) nor an interaction of administration with trial type (correct vs. incorrect) and/or SNR values on waiting time (GLM: $p = 1.25 \times 10^{-66}$; $\beta_{\text{administration}} = 0.12, p = 0.93$; $\beta_{\text{administration} \times \text{trial type}} = 0.30, p = 0.89$; $\beta_{\text{administration} \times \text{ratio}} = -0.21, p = 0.66$; $\beta_{\text{administration} \times \text{trial type} \times \text{ratio}} = 0.13, p = 0.84$; **Fig. 1A** below). In addition, we did not find a significant effect of drug administration type or interactions of administration type with trial type and SNR on reaction time ($p = 3.63 \times 10^{-73}$; $\beta_{\text{administration}} = -0.01, p = 0.86$; $\beta_{\text{administration} \times \text{trial type}} = 0.01, p = 0.91$; $\beta_{\text{administration} \times \text{ratio}} = 0.003, p = 0.91$; $\beta_{\text{administration} \times \text{trial type} \times \text{ratio}} = -0.002, p = 0.94$; **Fig. 1B** below).

We also analyzed the effect of drug administration type on the performance measures of probability of correct response, d' , and meta- d'/d' . We observed no significant effect of administration type or interaction with SNR on probability of correct response (GLM: $p = 1.64 \times 10^{-20}$; $\beta_{\text{administration}} = 0.003, p = 0.91$; $\beta_{\text{administration} \times \text{ratio}} = -0.003, p = 0.60$; **Fig. 1C** below). Furthermore, we found no significant effect of administration type or interaction of administration type with SNR values on d' (GLM: $p = 1.7 \times$

10^{-46} ; $\beta_{\text{administration}} = 0.40, p = 0.54$; $\beta_{\text{administration} \times \text{ratio}} = -0.18, p = 0.38$; **Fig. 1D** below). Finally, we did not observe a significant effect of administration type or interaction with ratio on meta- d'/d' (GLM: $p = 0.43$; $\beta_{\text{administration}} = 0.04, p = 0.97$; $\beta_{\text{administration} \times \text{ratio}} = -0.11, p = 0.74$; **Fig. 1E** below).

Together, these results do not provide any evidence that CNO (or its metabolic byproducts) act in a DREADDs-independent manner to account for the effects we report in our manuscript.

Figure 1. CNO Effects depend on DREADDs. (A) CNO had no significant effect on waiting time in the absence of DREADDs. Plotted are the distributions of the session-based average waiting times in EGFP sessions following CNO administration in orange and vehicle administration in yellow. The solid lines show the mean of the distributions. (B) CNO had no significant effect on reaction time in the absence of DREADDs. The plot is similar to panel A but for reaction time. Conventions are similar to panel A. (C) CNO had no significant effect on performance in the absence of DREADDs. The plot is similar to panel A but for the probability of correct response. Conventions are similar to panel A. (D) The type of administration in EGFP sessions had no significant effect on d' . Plotted is the d' for vehicle (yellow) and CNO (orange) administration. Error bars show S.E.M. (E) Similar to panel D but for meta- d'/d' . Conventions are similar to panel D.

Reviewers' Comments:

Reviewer #3:

Remarks to the Author:

The authors have addressed my remaining concern. I congratulate them on an interesting manuscript.